# Adapting non-invasive human recordings along multiple task-axes shows unfolding of spontaneous and over-trained choice

Yu Takagi[1,2,3]*, Laurence Tudor Hunt[2,4], Mark W Woolrich[2,4], Timothy EJ Behrens[4,5†], Miriam C Klein-Flügge[1,4†]*

[1]Wellcome Centre for Integrative Neuroimaging (WIN), Department of Experimental Psychology, University of Oxford, Oxford, United Kingdom; [2]Department of Psychiatry, University of Oxford, Warneford Hospital, Oxford, United Kingdom; [3]Department of Neuropsychiatry, Graduate School of Medicine, University of Tokyo, Tokyo, Japan; [4]Wellcome Centre for Integrative Neuroimaging (WIN), Centre for Functional MRI of the Brain (FMRIB), University of Oxford, Nuffield Department of Clinical Neurosciences, John Radcliffe Hospital, Oxford, United Kingdom; [5]Wellcome Trust Centre for Neuroimaging, UCL Institute of Neurology, University College London (UCL), London, United Kingdom

*For correspondence:
yutakagi322@gmail.com (YT);
miriam.klein-flugge@psy.ox.ac.uk
(MCK-F)

†These authors contributed
equally to this work

Competing interest: See
page 22

Reviewing editor: J Matias
Palva, University of Helsinki,
Finland

**Abstract** Choices rely on a transformation of sensory inputs into motor responses. Using invasive single neuron recordings, the evolution of a choice process has been tracked by projecting population neural responses into state spaces. Here, we develop an approach that allows us to recover similar trajectories on a millisecond timescale in non-invasive human recordings. We selectively suppress activity related to three task-axes, relevant and irrelevant sensory inputs and response direction, in magnetoencephalography data acquired during context-dependent choices. Recordings from premotor cortex show a progression from processing sensory input to processing the response. In contrast to previous macaque recordings, information related to choice-irrelevant features is represented more weakly than choice-relevant sensory information. To test whether this mechanistic difference between species is caused by extensive over-training common in non-human primate studies, we trained humans on >20,000 trials of the task. Choice-irrelevant features were still weaker than relevant features in premotor cortex after over-training.

## Introduction

For many decades, neuroscientists have studied task-dependent response properties of individual neurons, and this has laid the groundwork for our current understanding of brain function. More recently, a major shift from looking at individual neurons to studying population responses has begun to shed light on larger-scale neural dynamics, thus providing insight into previously hidden circuit mechanisms (*Cunningham et al., 2014*; *Yuste, 2015*). Usually, in this approach, the major axes of variation of a neural population are defined and the population activity is projected into this neural state space. This new way of looking at neural firing rates has begun to revolutionize our understanding of various neural processes, including the evolution of choice in parietal and frontal cortices (*Harvey et al., 2012*; *Mante et al., 2013*; *Morcos and Harvey, 2016*; *Raposo et al., 2014*), the critical stages of movement preparation and reaching in premotor and motor cortices (*Churchland et al., 2012*; *Kaufman et al., 2014*; *Li et al., 2016*), and the mechanisms underlying working memory in prefrontal cortex (PFC) (*Murray et al., 2017*).

Yet deriving neural population trajectories requires invasive recordings because the population vector is constructed from the firing rate of individual neurons. As a consequence, to date, no comparable non-invasive techniques for use in healthy human participants have been established. Here we asked whether we can recover 'population-like' trajectories in humans and thus provide insight into the unfolding of choice on a millisecond basis, by selectively suppressing neural populations to different features using repetition suppression in magnetoencephalography (MEG) recordings.

Repetition suppression (or 'adaptation') takes advantage of a feature first observed over 50 years ago (*Gross et al., 1967*), showing that the activity of a neuron will be suppressed when it is repeatedly exposed to features it is sensitive to. This phenomenon has since been widely replicated and shown to be a robust property of neurons in single-unit recordings (for review, see *Barron et al., 2016a*). Importantly, the bulk signal measured from thousands or millions of neurons using techniques such as electroencephalography (EEG), MEG, or functional magnetic resonance imaging (fMRI) will also show suppression if a subset of these neurons is sensitive to a repeated feature. Thus, repetition suppression provides insight into the activity of specific subpopulations of neurons in these non-invasive recordings available for use in humans and, in that way, allows us to examine the underlying neural mechanisms.

In this study, we focus on choice processes unfolding in dorsal premotor cortex (PMd). PMd is the key region for choosing hand digit responses (*Dum and Strick, 2002*; *Rushworth et al., 2003*), the response modality in our choice task. Thus, the neural representations of interest are located within one brain region. Given this spatial focus, repetition suppression provides the best resolution achievable using non-invasive MEG: a single sensor or voxel is sufficient to reveal feature-processing using repetition suppression. By contrast, multivariate approaches rely on spatial patterns detected across sensors, which would not offer the required spatial scale.

Here, we extend the repetition suppression framework in one crucial way: we suppress the MEG signal to multiple different features within the same experiment. Adaptation along each feature can be conceptualised as 'squashing' the neural response along one task dimension, or task-axis. This assimilates task-axes in multi-dimensional state space derived from recording many neurons, but using an experimental manipulation of a univariate, rather than multivariate, signal. Thus, we ask whether repetition suppression along multiple features can mimic projections onto multiple task-axes. If so, this would be the closest we can get to measuring multiple cellular-level representations within a single brain region in humans using MEG, with temporal resolution in the order of milliseconds, thanks to the temporal precision of MEG.

Our first key result shows that it is possible, using repetition suppression, to simultaneously induce adaptation along multiple task-axes in non-invasive human recordings. The recovered human choice traces show a progression from a processing of choice inputs to a processing of the motor response, just like in invasive recordings from non-human primates (NHPs) (*Mante et al., 2013*).

Once the feasibility of our non-invasive adaptation approach was established, we asked whether the mechanisms of choice uncovered in NHPs also generalize to humans. More precisely, we examined whether the mechanisms for input selection during choice were comparable between the two species. A large body of evidence in human and non-human primates shows that selection of relevant sensory inputs occurs through top-down modulations from prefrontal and parietal regions onto early sensory regions (*Buschman and Miller, 2007*; *de Lange et al., 2010*; *Desimone and Duncan, 1995*; *Egner and Hirsch, 2005*; *Everling et al., 2002*; *Friston, 2005*; *Friston and Kiebel, 2009*; *Gazzaley et al., 2007*; *Kok et al., 2016*; *Michalareas et al., 2016*; *Moore and Zirnsak, 2017*; *Noudoost et al., 2010*; *Pezzulo and Cisek, 2016*; *Rao and Ballard, 1999*; *Richter et al., 2017*; *Squire et al., 2013*; *van Wassenhove et al., 2005*; *Wyart et al., 2015*). However, two recent perceptual decision-making studies in macaques found that irrelevant sensory inputs are not filtered out before the integration stage (*Mante et al., 2013*; *Siegel et al., 2015*). To reconcile these findings, naive human participants performed context-dependent choices in the same perceptual decision-making task used in macaques. This showed that information about both relevant and irrelevant input dimensions was present in premotor cortex, but irrelevant inputs were weaker than relevant inputs, consistent with top-down suppression.

One major difference in terms of how human and animal studies are conducted, however, is that animals are trained for thousands of trials before neural recordings are performed. Differences in input selection mechanisms could therefore be a consequence of brain circuit reorganization caused by extensive over-training. In an attempt to reconcile our findings with those obtained in macaques,

we next mirrored NHP training conditions, and our human participants underwent extensive training on over 20,000 trials of the task. However, this did not change the input selection mechanisms evident in the MEG recordings taken afterwards. Irrelevant inputs were present, but still more weakly than relevant inputs. This was true when examining information processed in premotor cortex, or when decoding from whole-brain activity.

## Results

Participants (n = 22) performed a modified version of a dynamic random-dot-motion (RDM) decision task, in which stimuli simultaneously contained information about colour (red vs. green) and motion (left vs. right, as in *Mante et al., 2013*). An additional flanker stimulus displayed 150 ms before the RDM stimulus instructed participants about the relevant stimulus dimension: arrows indicated the choice in the current trial was about the dominant motion direction, whereas coloured dots instructed participants to respond based on the dominant colour (*Figure 1A*).

### MEG activity can track the temporal evolution of a choice process

When recording activity from premotor cortex in NHPs performing the same task, neural activity shows a clear evolution from an initial processing of the choice input (e.g., colour in a colour trial) to a later processing of the choice direction (motor response: right or left; *Mante et al., 2013*). Our first analysis, therefore, aimed to establish whether the time-resolved nature of MEG data would allow us, in a similar way, to watch how a choice evolved from the processing of inputs to the processing of the motor response in human premotor cortex.

In order to get a handle on neurons that process colour, motion, or response in MEG recordings measuring the summed activity of many neurons, we incorporated an additional task manipulation. The main RDM stimulus on each trial ('test stimulus': TS) was preceded by another RDM stimulus, the 'adaptation stimulus' (AS), so that each trial contained two RDM stimuli in quick succession (*Figure 1A*). Unlike the TS, the AS contained only one input feature (strong red or strong green colour, or strong leftward or strong rightward motion).

Participants responded to both stimuli using the corresponding finger of the right hand (index finger for leftward motion or green colour, middle finger for rightward motion or red colour). We hypothesized that presentation of the AS meant that MEG activity recorded at the time of the TS would have suppressed responses to features already engaged at the time of the AS. For example, when a red and leftward TS was preceded by a red AS, MEG activity measured at the time of the TS would still contain the activity of neurons responding to leftward motion, but neurons responding to the red colour input would be suppressed, compared to a situation where the AS was green (*Figure 1B*). Furthermore, if participants were asked to attend to the colour of the TS, they would produce a middle finger response twice in quick succession (to AS and TS, both red), and thus the 'response-direction (middle finger)' selective neurons would also be suppressed (*Figure 1C*). This would not be the case for the identical stimuli when leftward motion was attended at the time of the test stimulus. In this case, only colour-sensitive neurons but not response-selective neurons would be selectively suppressed at the time of the TS. In summary, by creating and comparing situations with and without suppression for input (colour or motion) and response, we aimed to establish whether we could measure premotor cortex activity transition from a processing of input to a processing of choice output using non-invasive human MEG.

We focused this analysis on the time-course of activity from an area in left premotor cortex (PMd). PMd was identified by first projecting the MEG scalp signal back into source space, which relies on linearly combining MEG sensors to create 'virtual sensors' reflecting the signal from specific cortical locations (linearly constrained minimum variance (LCMV)-beamforming; see 'Materials and methods'). At the source level, we then computed a contrast between TS trials with repetition suppression to any input (colour or motion) or response at any time point between $[-250,750]$ ms around the test stimulus and TS trials containing no such suppression (see 'Materials and methods'). This comparison made use of all trials in the experiment and matched the two sides of the contrast in terms of the visual stimulus properties of the TS and the required motor response. The only difference between the two sides of the contrast was which AS preceded a given TS and whether, as a result, we expected repetition suppression to a shared feature or not. Consequently, this contrast was agnostic to any differences between input and response suppression, any

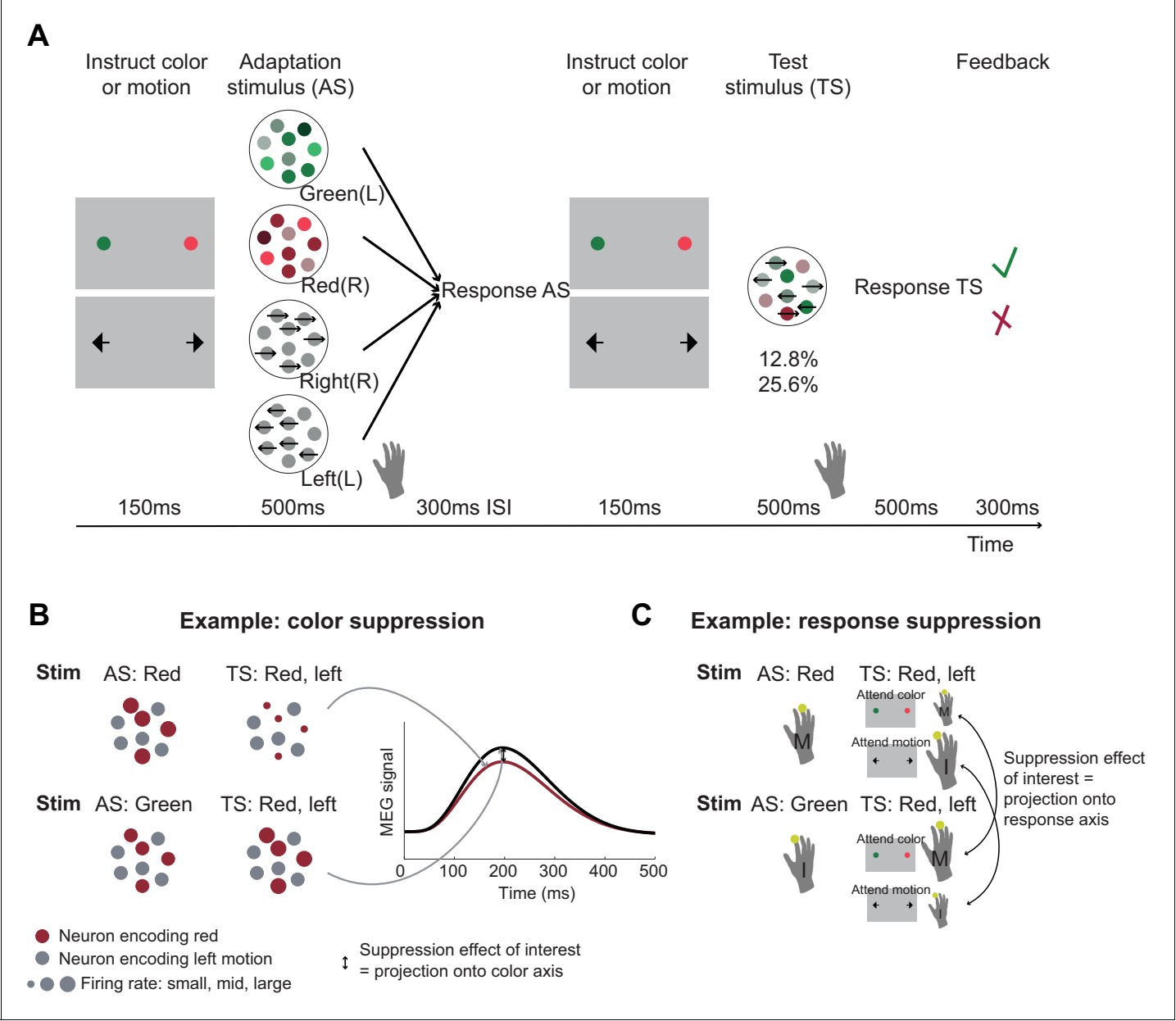

**Figure 1.** Experimental task involving manipulation of specific choice features. (**A**) Human participants performed a perceptual choice task adapted from the macaque version in *Mante et al., 2013* while magnetoencephalography (MEG) data were recorded. Each trial involved two random-dot-motion stimuli – an adaptation stimulus (AS) and a test stimulus (TS). A flanker cue (coloured dots or arrows) instructed which choice dimension, colour or motion, to attend to for making a choice. Responses were given using the right-hand index or middle finger for green/left and red/right stimuli, respectively. By presenting two choices with varying features in rapid succession, we selectively suppressed the subset of neurons sensitive to repeated features. To maximize suppression effects, AS colour and motion was strong compared to the TS (70% compared to 25.6 or 12.8% motion coherence or colour dominance). Feedback at the end of each trial related to performance on the TS. In total, there were 64 conditions: 4 AS x 2 contexts x 2 directions x 2 colours x 2 coherence levels. The rationale for the selective suppression of choice features is illustrated for two examples in (**B**) and (**C**). (**B**) The top and bottom rows show two combinations of AS and TS that were compared to extract colour suppression. At the time of the TS (the focus of all analyses), the stimulus is identical, containing predominantly red colour and left-ward motion. If preceded by a red AS (top row), any red-coding neurons will show a reduced signal at the time of the TS (red dots), but any other neurons will show the same response (grey dots). Thus, the overall MEG signal will be reduced compared to a situation where the preceding AS stimulus does not share any features with the TS (green; bottom row). This suppression effect, i.e., the difference in the MEG signal for two identical TS as a function of their preceding AS (arrow) can be captured in a time-resolved manner, thus showing not only whether but also *when* colour is being processed. This experimental repetition suppression manipulation can be conceptualized as a projection (or 'squashing') of the MEG signal onto the axis that captures variation in colour processing. (**C**) The sequence of stimuli shown in (**B**) can also be used to probe response suppression. When participants are attending to colour at the time of the TS, a middle finger

*Figure 1 continued on next page*

*Figure 1 continued*

response will be repeated in the top example but not in the bottom. If they are attending to motion, an index finger response will be repeated in the bottom but not in the top example. The respective differences (arrows) thus provide a time-resolved measure of response suppression, analogous to projections of neurons onto a response axis.

differences between relevant and irrelevant input suppression, and any timing differences between these effects. PMd, the premotor region responsible for selecting hand motor responses, was our a priori region of interest for this study. Its role in selecting finger responses is equivalent to the role of the frontal eye fields (FEF) for guiding eye movement responses, the region where NHP recordings were performed (*Mante et al., 2013*). Indeed, left PMd, in a cluster together with left M1, was the strongest peak at the whole-brain level for this contrast (p(FWE)=0.018 cluster-level corrected; left M1: z = 6.10, peak MNI coordinate x=-36, y=-18, z = 48; left PMd: z = 5.02, peak MNI coordinate x=-37, y=-6, z = 55; right PMd: z = 3.11 at x = 37, y = 06, z = 55; *Figure 2A*). A further analysis performed on a 38-region parcellation using beamforming with orthogonalization (*Colclough et al., 2016*; *Colclough et al., 2015*) confirmed that the linear regression coefficients for both input and response suppression were strongest in the parcel containing premotor cortex (*Figure 2—figure supplement 1*).

To examine how the choice evolved over time in PMd, we extracted the timeseries from left PMd (x=-37, y=-6, z = 55) and used an L2-regularized linear regression (ridge regression) containing regressors that described which task variables were being suppressed on a trial-by-trial basis (see 'Materials and methods'). The regression was applied to each time point around the presentation of the TS ([−500,1200] ms) using a sliding-window approach (window size: 150 ms). Simulations showed that all regressors were estimable despite dependencies between some (maximum shared variance: $r^2 = 0.34$; *Figure 2—figure supplements 2* and *3*). *Figure 2B* demonstrates the temporal evolution of the choice for each of the regressors. The two contextual variables 'context' (motion or colour instruction flankers) and 'switch' (attended dimension same or different from AS) showed the earliest significant representation, and this response was sustained throughout the TS presentation (both first significant at 8 ms). This is unsurprising as contextual information was available −150 ms before the TS. Both of these variables were unrelated to the repetition suppression manipulation. Slightly later, starting from 108 ms, the input representation emerged (probing whether or not the same input feature was already shown in the AS and thus suppressed). The regressor that was represented latest in PMd was the motor response (probing whether the same finger was/was not used to respond to the AS and thus suppressed) and the choice direction (left or right; both starting from 275 ms). Input and response regressors infer sensory and motor processing using the repetition suppression manipulation; choice direction, by contrast, simply captures the TS response direction, independent of any repetition.

Statistical tests performed on the peaks of the linear regression coefficients obtained for the five regressors showed a significant difference in processing latency (one-way repeated-measures ANOVA: $F(4, 84)=128.71$, p=5.5154e-35, $\eta^2_p = 0.86$; *Figure 2C*), and a pairwise post-hoc test between the two critical regressors, input and response repetition suppression, confirmed a significant timing difference with the choice input representations preceding the response representations (pairwise *t*-test on peak linear regression coefficients for input [429.5 ± 35 ms; mean ± s.t.d.] and response [531.1 ± 41.6 ms]: $t(21)=11.07$, p=3.1654e-9, after Bonferroni familywise error correction; Hedges' $g = 2.577$, 95% CI = [1.8647, 3.4166]; see 'Materials and methods'; *Figure 2C*). Thus, activity recorded using MEG repetition suppression in human premotor cortex can track the evolution of a choice from sensory input to motor response.

This transition from input-to-choice processing can also be plotted as a continuous trace that progresses along the two task-axes. This is achieved by plotting the linear regression coefficients capturing input suppression, as a marker of input processing, on the first axis and the linear regression coefficients capturing response suppression, as a marker of motor response processing, on the second axis (*Figure 3A*). The obtained choice trace mimics the state space trajectories derived from neuronal population responses in previous NHP recordings (*Mante et al., 2013*). This way of plotting the data does not add information but helps visualize the choice dynamics in the way usually done when projecting neural populations into state space. A key difference, however, is that our approach

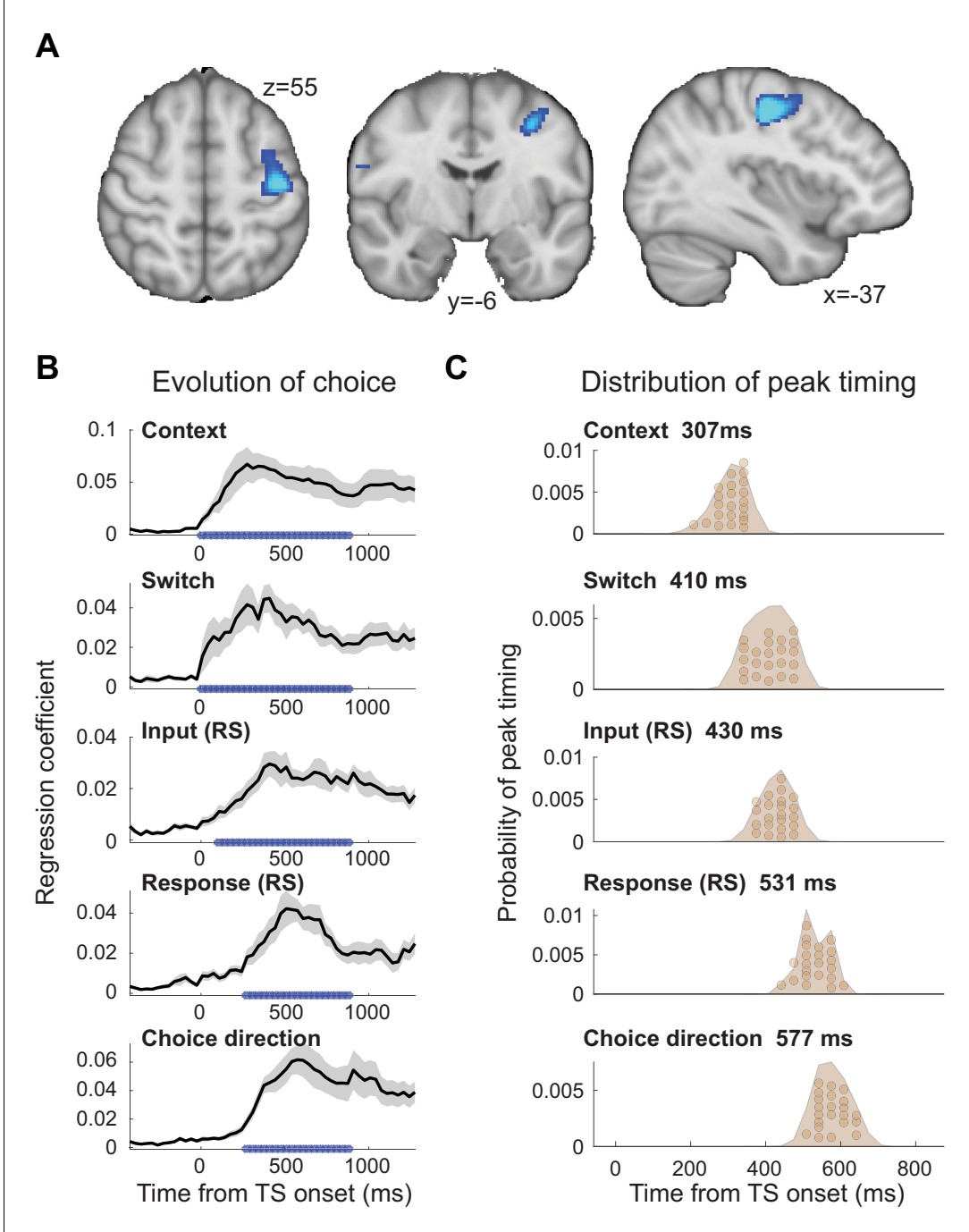

**Figure 2.** The evolution of choice in human premotor cortex. (**A**) Dorsal premotor cortex (PMd) was the a priori region of interest for this study. Indeed, using source localization and a contrast between trials containing any input or response adaptation, compared to no adaptation, revealed a cluster involving M1 and left PMd (group contrast, whole-brain cluster-level FWE-corrected at p<0.05 after initial thresholding at p<0.001; PMd peak coordinate: x=-37, y=-6, z = 55). Importantly, this contrast was agnostic and orthogonal to any differences between input and response adaptation, relevant and irrelevant input adaptation, or pre- vs. post-training effects (including any timing differences, as beamforming was performed in the time window [−250, 750] ms). (**B**) The time-resolved nature of magnetoencephalography (MEG), combined with a selective suppression of different choice features, allowed us to track the evolution of the choice on a millisecond timescale. Linear regression coefficients from a regression performed on data from left premotor cortex (PMd) demonstrate an early representation of context (motion or colour) and switch (attended dimension same or different from TS) from ~8 ms (sliding window centred on 8 ms contains 150 ms data from [−67,83] ms). The representation of inputs, as indexed using repetition suppression (RS), emerged from around 108 ms (whether or not the input feature, colour, or motion was repeated). Finally, the motor response, again indexed using RS (same finger used to respond to adaptation stimulus [AS] and test stimulus [TS] or not), and TS choice direction (left

*Figure 2 continued on next page*

*Figure 2 continued*

or right, unrelated to RS) were processed from 275 ms. *p<0.001; error bars denote SEM across participants; black line denotes group average. (C) The distribution of individual peak times across the 22 participants directly reflects this evolution of the choice process. In particular, it shows significant differences in the processing of input and response, consistent with premotor cortex transforming sensory inputs into a motor response.

The online version of this article includes the following source data and figure supplement(s) for figure 2:

**Source data 1.** Contains 'pre' and 'post' [Time x Regressors x Subjects] for *Figure 2B*.
**Source data 2.** Contains 'dat' [Regressors x Subjects] for *Figure 2B*.
**Figure supplement 1.** Premotor cortex processes choice inputs and outputs.
**Figure supplement 1—source data 1.** Contains 'pre' and 'post' [ROIs x Regressors].
**Figure supplement 2.** Simulations show sufficient parameter recovery given the experimental design.
**Figure supplement 2—source data 1.** Contains 'a_est', 'b_est', 'c_est' [Subjects x Regressors], and 'betas_true' [Regressors x 1].
**Figure supplement 3.** Simulations show sufficient independence between estimated linear regression coefficients.
**Figure supplement 3—source data 1.** Contains 'a', 'b' [Regressors x Regressors], 'c_betas_true' and 'c_betas_pred' [Repetition x Regressors x Subjects].

did not rely on invasively recorded firing rates but was possible using non-invasive repetition suppression along multiple features.

## Irrelevant inputs are processed less relative to relevant inputs in human PMd

Having established that different components of the choice computation can be tracked in a temporally resolved manner using MEG, we next examined whether PMd processes inputs equally when they are relevant compared to when they are irrelevant for the choice at hand. Accounts of top-down attentional filtering would predict reduced processing of inputs that are irrelevant for making

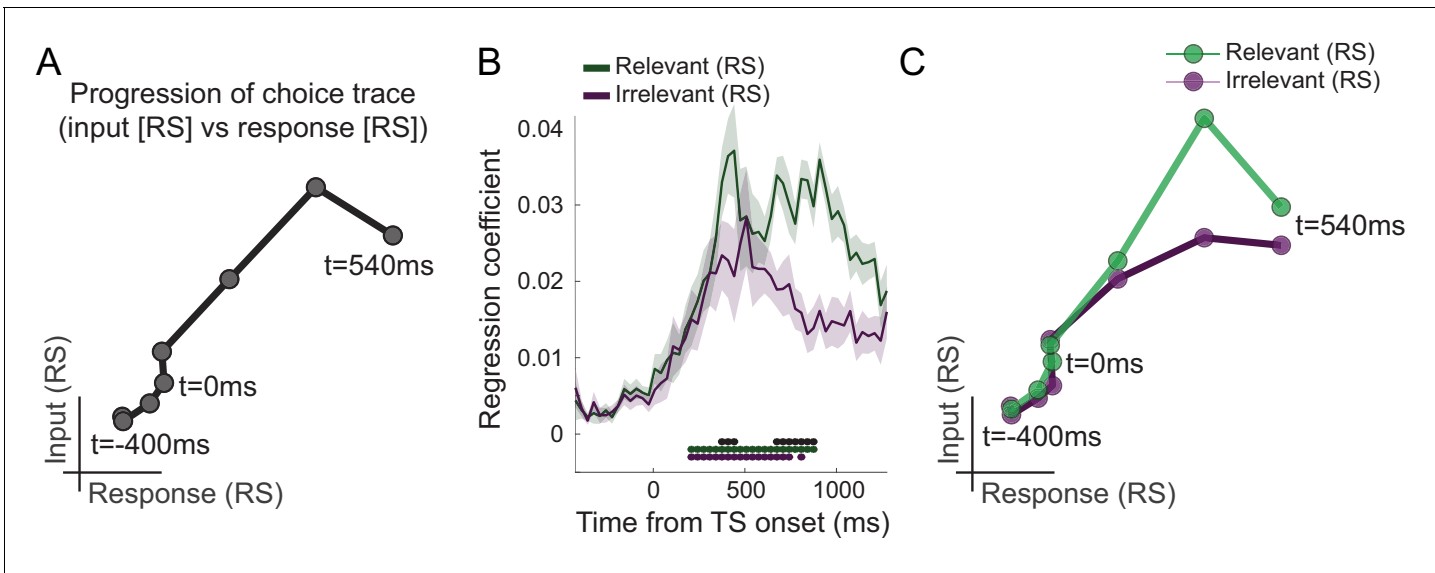

**Figure 3.** Choice trace: progression along task-axes shows filtering out of irrelevant inputs in PMd. (A) By plotting the processing of input on one and response on the other axis (both as in *Figure 2B*), we can derive a choice trace that shows the signal progression along the two task-axes, thus mimicking state space trajectories obtained from non-human primate (NHP) recordings (*Mante et al., 2013*). (B) Separation of input suppression into relevant and irrelevant inputs shows slightly diminished processing of irrelevant inputs in dorsal premotor cortex (PMd) (p<0.001; black * shows the difference between relevant and irrelevant inputs, significant between 375–442 and 675–875 ms; green and purple * indicates significance separately for relevant and irrelevant inputs). (C) This can also be seen in the choice trace split for relevant and irrelevant inputs. We observed partial but incomplete filtering of irrelevant inputs in PMd.

The online version of this article includes the following source data for figure 3:

**Source data 1.** Contains 'pre.relirrel' and 'post.relirrel' [Time x Regressors x Subjects].
**Source data 2.** Contains structs of 'pre' and 'post'.

a choice (e.g., colour when the choice is about motion). By contrast, recent evidence provided from recordings in NHPs suggests the passing forward of both relevant and irrelevant inputs, with selection occurring at the output stage (*Mante et al., 2013*). Comparing the linear regression coefficients capturing the processing of relevant and irrelevant inputs, both indexed using repetition suppression, showed that both were processed significantly from 208 ms, but importantly, when the suppressed sensory dimension was relevant to the choice at hand, this had a stronger effect on the signal compared to when it was irrelevant ($p < 0.001$ between 375–442 and 675–875 ms; *Figure 3B*). In other words, repetition suppression effects were greater for relevant compared to irrelevant sensory information, suggesting prioritized processing of choice-relevant over choice-irrelevant sensory inputs in PMd. This can also be appreciated in the alternative visualization as two-dimensional choice traces for relevant and irrelevant inputs (*Figure 3C*). Thus, our data show partial but incomplete filtering of irrelevant inputs in PMd. Feature selection did not occur solely at the motor output stage of the decision process, nor was irrelevant information entirely filtered out by this stage.

In trials with irrelevant input suppression, by design, participants had to switch context between AS and TS, for example, from attending colour to attending motion. Context switches, however, also sometimes occurred in the absence of irrelevant input suppression. Importantly, simulation analyses showed that linear regression coefficients could be estimated accurately despite some shared variance between switch and irrelevant input regressors (simulations; *Figure 2—figure supplements 2* and *3*).

## Over-training does not abolish the filtering out of irrelevant inputs

We speculated that one potential cause for seeing attenuated irrelevant sensory input processing in our human participants who spontaneously performed the task, but unattenuated irrelevant input processing in recordings from NHPs, may be rooted in the fact that monkeys have been trained to perform this task for many months and thousands of trials. This may have caused plasticity in the neural pathways that process and pass forward the input information. To test this hypothesis, we trained our human participants for 4 weeks on over 20,000 trials of the task (see 'Materials and methods'; *Figure 4—figure supplement 1A*). Unsurprisingly, this led to a speeding up of reaction times and higher performance scores ($2 \times 2 \times 3$ repeated-measures ANOVA with factors context [motion vs. colour], training [pre vs. post], and coherence [70% (AS) vs. 25.6% vs. 12.8%]; effect of training on log-reaction times (RTs): $F_{(1,20)}=9.23$, $p=6.46e-3$, $\eta^2_p = 0.32$, mean RT difference $29 \pm 10$ ms; effect of training on % correct: $F_{(1,20)}=36.39$, $p=6.75e-6$, $\eta^2_p = 0.65$, mean difference $4.76 \pm 0.8$; *Figure 4A* and *Figure 4—figure supplement 1B,C*). Upon completion of the training, we again recorded MEG data during performance of the task in the same way as done in pre-training. As before, the unfolding of the choice computation was evident in signals recorded from PMd (*Figure 4C–D*). Crucially, however, when we repeated the analysis focussing on any differences between the processing of relevant and irrelevant inputs, the difference was retained (*Figure 4E–G*). Even after having performed thousands of trials on the task, irrelevant inputs were, therefore, still filtered out from PMd activity. There was no interaction between input processing and training (two-way repeated-measures ANOVA with factors input [relevant vs. irrelevant] and training [pre vs. post]: all $p > 0.05$ for effect of input x training; Bonferroni correction for familywise error rate; a two-way repeated-measures Bayesian ANOVA provided further evidence in favour of the null hypothesis: the input x training interaction was not part of the winning model at any time point).

Consistent with faster behavioural reaction times, we did observe changes in peak processing latency of the different components of the choice process after completion of an extensive training regime (*Figure 4D*; two-way repeated-measures ANOVA with factors regressors [Context, Switch, Input repetition suppression, Response repetition suppression, Choice] and training [pre vs. post]: effect of regressors [$F_{(4,84)} = 39.35$, $p=1.5824e-18$, $\eta^2_p = 0.65$]; effect of training [$F_{(1,84)}=75.81$, $p=2.0638e-08$, $\eta^2_p = 0.78$]; effect of regressors x training [$F_{(4,84)}=35.38$, $p=2.6436e-17$, $\eta^2_p = 0.63$]). Post-hoc tests confirmed that peak timings changed for four out of five regressors following the training, including the processing of context switches, input adaptation, response adaptation, and choice direction (pairwise *t*-test between pre vs. post on the peak timings of context: $t(21) = 0.18$, $p=1.00$, Hedges' $g = 0.046$, 95% CI = [−0.481, 0.575]; switch: $t(21) = 5.92$, $p=3.5641e-05$, Hedges' $g = 1.98$, 95% CI = [−0.481, 0.575]; input repetition suppression: $t(21) = 5.91$, $p=3.6398e-05$, Hedges' $g = 1.57$, 95% CI = [0.94, 2.27]; response repetition suppression: $t(21) = 12.15$, $p=2.8914e-$

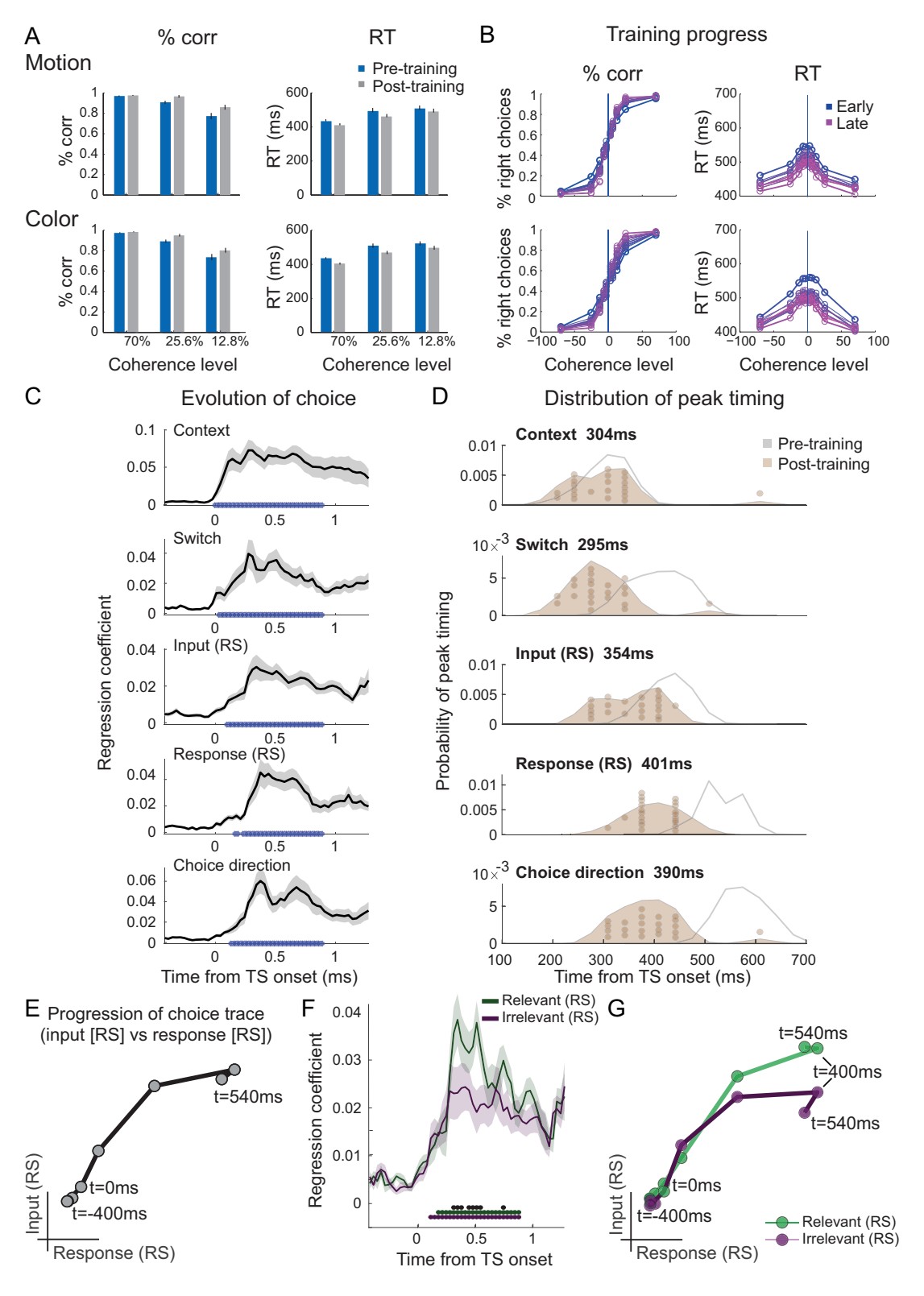

**Figure 4.** Filtering out of irrelevant inputs remains after over-training on >20,000 trials. (A) Over a period of 4 weeks, participants performed >1000 trials of training on the random-dot-motion task for 5 or 6 days a week before returning for two further post-training magnetoencephalography (MEG) sessions. Comparison of pre- and post-training task performance showed significant reaction time (RT) speeding and performance improvements (% correct choices) in both colour and motion contexts. (B) Performance improvements are also clear in psychometric curves plotted across training

*Figure 4 continued on next page*

*Figure 4 continued*

sessions (blue = early, purple = late; see also *Figure 5A* and *Figure 5—figure supplement 1A*). (C) The evolution of a choice can be tracked in PMd post-training (as shown in *Figure 2B* for the pre-training sessions). (D) However, peak timings of the majority of regressors occur earlier in the post-compared to the pre-training data, suggesting a faster and possibly more efficient coding of the choice process. (E) The choice trace extracted from the repetition suppression regressors for input and response in PMd shows a transformation from a processing of inputs to a processing of outputs, similar to *Figure 3A*, for the data acquired pre-training. (F) The difference between the processing of relevant versus irrelevant inputs is preserved even after extensive training, thus suggesting some filtering out of irrelevant inputs. (G) This can be visualized in two divergent choice traces, separately showing relevant and irrelevant input adaptation.

The online version of this article includes the following source data and figure supplement(s) for figure 4:

**Source data 1.** Contains 'dat' [Regressors x Subjects].
**Figure supplement 1.** Timeline, performance progression during training sessions, and behavioural effects of relevant and irrelevant inputs and AS.
**Figure supplement 2.** Transition from choice to response processing is specific to PMd.
**Figure supplement 2—source data 1.** Contains 'pre' and 'post' [Time x Regressors x Subjects] for LIP.
**Figure supplement 2—source data 2.** Contains 'pre' and 'post' [Time x Regressors x Subjects] for PMd.
**Figure supplement 2—source data 3.** Contains 'pre' and 'post' [Time x Regressors x Subjects] for V1.
**Figure supplement 2—source data 4.** Contains 'pre' and 'post' [Time x Regressors x Subjects] for V4.
**Figure supplement 2—source data 5.** Contains 'pre' and 'post' [Time x Regressors x Subjects] for mPFC.
**Figure supplement 3.** Task feature processing independent of repetition suppression in PMd.
**Figure supplement 3—source data 1.** Contains 'pre' and 'post' [Time x Regressors x Subjects].

10, Hedges' $g$ = 3.08, 95% CI = [2.28, 4.02]; choice: $t(21)$ = 11.09, p=1.5246e-09, Hedges' $g$ = 3.28, 95% CI = [2.35, 4.37]; Bonferroni correction for familywise error rate; *Figure 4D*).

## Behavioural effects predicted by neural adaptation mechanisms

We have shown that repeated exposure to sensory inputs or motor responses has an impact on the MEG signal recorded at the time of the TS. However, if repetition has such a clear effect on neural representations, it is possible that it might also impact participants' behaviour. While the precise neuronal mechanisms underlying repetition suppression are not fully understood, one possibility is that a fatigue mechanism attenuates the inputs of the repeated feature (*Barron et al., 2016a*; *Grill-Spector et al., 2006*; *Vidyasagar et al., 2010*). This would affect the neural information content that is being processed and might translate into behavioural change.

To test this, we repeated the regression conducted at every time step in PMd on log-reaction times (RTs) and response accuracies (% correct) of TS choices (see 'Materials and methods'). We hypothesized that suppression of a relevant sensory feature or motor representation would slow RTs and reduce accuracy. For example, if a green AS meant that green inputs were attenuated at the time of the TS, we would predict TS reaction times and accuracy to be impacted when green is the relevant TS feature to attend to.

Consistent with our predictions, participants were slower and less accurate when the relevant input had already been processed at the time of the AS and was therefore suppressed at the time of the TS (RT: $t(21)$=6.80, p=1.00e-6, Hedges' $g$ = 2.00, 95% CI = [1.18, 2.92]; accuracy: $t(21)$=-8.65, p=2.28e-8, Hedges' $g$ = −2.55, 95% CI = [−3.60,−1.62]). Similarly, they were less accurate when the motor response was repeated (accuracy; $t(21)$=-2.28, p=0.03, Hedges' $g$ = −0.67, 95% CI = [−1.33,−0.05]; *Figure 5*; see *Figure 5—figure supplement 1* for separated results of pre- and post-training), akin to choice history biases that tend to prefer alternating choices (*Urai et al., 2019*). However, neither RTs nor choice accuracies were affected by repetition of the irrelevant sensory feature (RT: $t(21)$=0.13, p=0.90, Hedges' $g$ = 0.04, 95% CI = [−0.57, 0.64]; accuracy: $t(21)$=0.03, p=0.98, Hedges' $g$ = 0.01, 95% CI = [−0.60, 0.61]). Other main effects included context (RT: $t(21)$=-2.31, p=0.03, Hedges' $g$ = −0.68, 95% CI = [−1.34,−0.06]; accuracy: $t(21)$=2.48, p=0.02, Hedges' $g$ = 0.73, 95% CI = [0.10, 1.39]) and context-switch (RT: $t(21)$=7.72, p=1.46e-7, Hedges' $g$ = 2.27, 95% CI = [1.40, 3.25]; accuracy: $t(21)$=-9.58, p=4.10e-9, Hedges' $g$ = −2.82, 95% CI = [−3.95,−1.82]).

## Task feature processing independent of repetition suppression in PMd

The results described thus far have relied on the use of repetition suppression to manipulate the MEG signals recorded at the time of the TS. As outlined in 'Introduction', this is what our experiment was optimized for. Suppression along multiple task features allows us to measure multiple cellular-

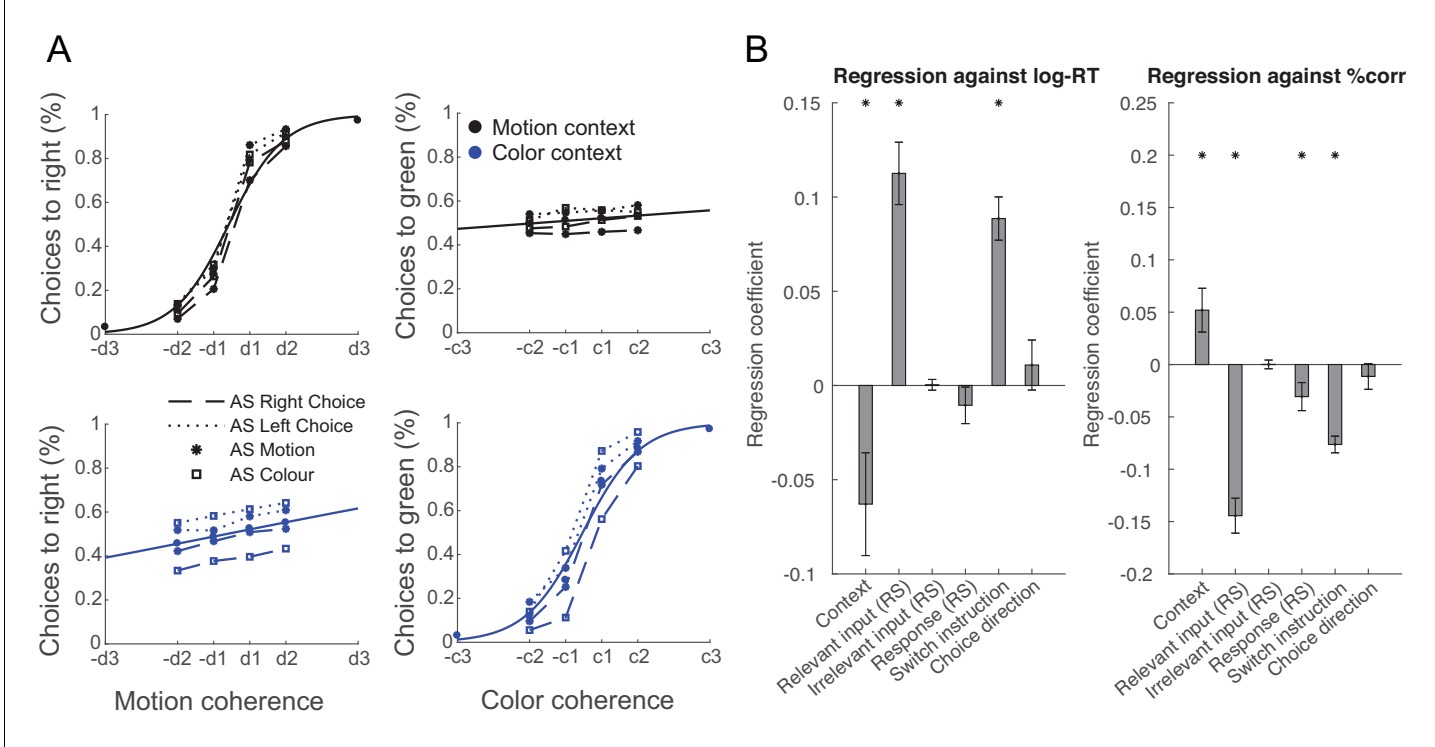

**Figure 5.** Manipulating information content through repetition suppression changes behaviour. (**A**) Psychometric curves show a strong influence of the relevant feature (colour in a colour trial and motion in a motion trial) and a weak influence of the irrelevant feature (colour in a motion trial and motion in a colour trial), consistent with data in non-human primates (*Mante et al., 2013*). In addition, suppressing motion inputs affects choices based on motion and suppressing colour inputs affects choices based on colour, but not vice versa. (**B**) Relevant, but not irrelevant, input adaptation slowed log-reaction times (RTs), and both relevant input adaptation and response adaptation reduced choice accuracies (% correct). This is consistent with a potential neuronal fatigue mechanism whereby repeated exposure to a feature reduces the received input, which would be expected to affect behaviour.

The online version of this article includes the following figure supplement(s) for figure 5:

**Figure supplement 1.** Pre- vs. post-training effects of repetition suppression on behavioural performance.

level representations within the same brain region. By contrast, multivariate approaches akin to those used in *Mante et al., 2013* would in humans rely on spatial patterns detected across MEG sensors, which would have a lower spatial resolution, not allowing us to examine processes within one brain region.

Nevertheless, across trials, we can ask whether univariate signal variance in PMd relates to task features such as sensory input strength or choice direction using an encoding-style regression analysis, which mimics the approach used in *Mante et al., 2013* (for details, see 'Materials and methods'). This is not an optimal analysis approach, given our experimental design (see 'Materials and methods'), and, unlike our repetition suppression (RS) approach, it may capture confounding variables (e.g., attention, difficulty). Still, the results are broadly consistent with our original findings, showing that PMd's univariate signal contains information related to all task features (context, sensory input strength, choice direction). The strength of sensory inputs is processed earlier in time than the choice direction (motor response), and the strength of irrelevant inputs is processed more weakly than the strength of relevant inputs both before and after extensive training (*Figure 4— figure supplement 3*).

## Spatial specificity of PMd effects

PMd was our a priori region of interest for this study and the strongest peak at the whole-brain level (contrast probing input and response adaptation). To further show the specificity of our findings, we repeated all key analyses for a few other a priori regions known to be implicated in visual feature

selection and choice: primary visual cortex (V1), higher-order visual cortices (V4 and V5), medial pre-frontal cortex (mPFC), and lateral intraparietal area (LIP). None of these regions showed the transition from processing choice input to processing the motor response observed in PMd (*Figure 2—figure supplement 1* for all regions; *Figure 4—figure supplement 2* for a priori regions).

## Filtering out of irrelevant inputs is present across the brain

Finally, it is possible that our targeted focus on PMd as the 'output' stage of the decision process might have obscured the fact that other MEG sensors processed irrelevant input information more strongly. To investigate this, we performed a multivariate decoding analysis on the data from 38 parcels covering the whole brain in source space. This analysis focused on stimulus properties of the TS, disregarding any repetition suppression. It also accounted for shared variance between relevant sensory inputs and choice direction by working on the residuals of the MEG signal after regressing out variance related to choice direction (see 'Materials and methods' for details). Separate decoders were constructed to decode relevant and irrelevant sensory input strength (coded as $[-2, -1, 1, 2]$) from these residuals. Consistent with earlier analyses, decoding across all parcels showed weaker decoding of irrelevant compared to relevant inputs (*Figure 6*). There was some significant decoding of irrelevant inputs (p<0.001 from 455 to 708.3 ms post-TS) but it was again less extended in time and weaker compared to the decoding of relevant inputs (p<0.001 from 201.7 to 961.7 ms post-TS; *Figure 6A*). There was again no difference between the data acquired pre- vs. post-training (two-way repeated-measures ANOVA with factors input [relevant vs. irrelevant] and training [pre vs. post]: all p>0.5; minimum p=0.52, $F(1,21) = 5.49$, $\eta^2_p = 0.21$, for effect of training; Bonferroni correction for familywise error rate; a two-way repeated-measures Bayesian ANOVA provided further evidence in favour of the null hypothesis: the input x training interaction was not part of the winning model at any time point) apart from differences in peak timing (*Figure 6B*; pairwise *t*-test between pre vs. post on the peak timings of input repetition suppression: $t(21) = 3.12$, p=0.026, Hedges' $g = 1.00$, 95% CI = [0.33, 1.73]; context: $t(21) = 2.35$, p=0.14, Hedges' $g = 0.62$, 95% CI = [0.08, 1.20]; switch: $t(21) = -0.60$, p=1.00, Hedges' $g = -0.19$, 95% CI = [−0.86, 0.46]; response repetition suppression: $t(21) = 1.59$, p=0.63, Hedges' $g = 0.52$, 95% CI = [−0.15, 1.22]; choice: $t(21) = 2.30$, p=0.16, Hedges' $g = 0.67$, 95% CI = [0.07, 1.30]), as observed in PMd (*Figure 4D*). This further supports an interpretation of increased efficiency as a result of extensive over-training. Thus, while this analysis cannot rule out that there may be a brain region that represents irrelevant inputs more strongly, we confirmed that even after extensive training and when considering the activity across the whole brain, the processing of irrelevant inputs was attenuated compared to the processing of relevant inputs when making a choice.

## Discussion

Here we have shown that non-invasively recorded MEG activity can be used to track a choice process on a millisecond timescale. As decisions unfolded, MEG activity in premotor cortex transitioned from processing sensory inputs to processing the motor response. Watching these dynamics has previously only been possible by projecting high-dimensional neural population recordings onto a low-dimensional set of axes or 'neural state space'. Here, we took a different approach to assimilate this state space, in which we selectively suppressed the processing of sensory inputs or response features in the average neural data recorded with MEG. This allowed us to selectively index (or 'squash') neural activity along the different task-axes that would define the neural state space in direct neuronal recordings. We found that in human premotor cortex, sensory inputs irrelevant to the current choice were processed more weakly compared to relevant choice inputs. This partial filtering of irrelevant choice features was observed even after extensive over-training and when considering activity present across the brain.

Studying neural population responses, as opposed to the responses of individual neurons, has received increasing attention because it provides a window into larger-scale neural dynamics (*Yuste, 2015*). It has provided crucial insights, for example, into the evolution of choice (*Harvey et al., 2012*; *Hunt et al., 2018*; *Mante et al., 2013*; *Morcos and Harvey, 2016*; *Raposo et al., 2014*), movement preparation and execution (*Churchland et al., 2012*; *Kaufman et al., 2014*; *Li et al., 2016*), and the mechanisms underlying working memory (*Murray et al., 2017*). The approach we used to derive representations linked to multiple features,

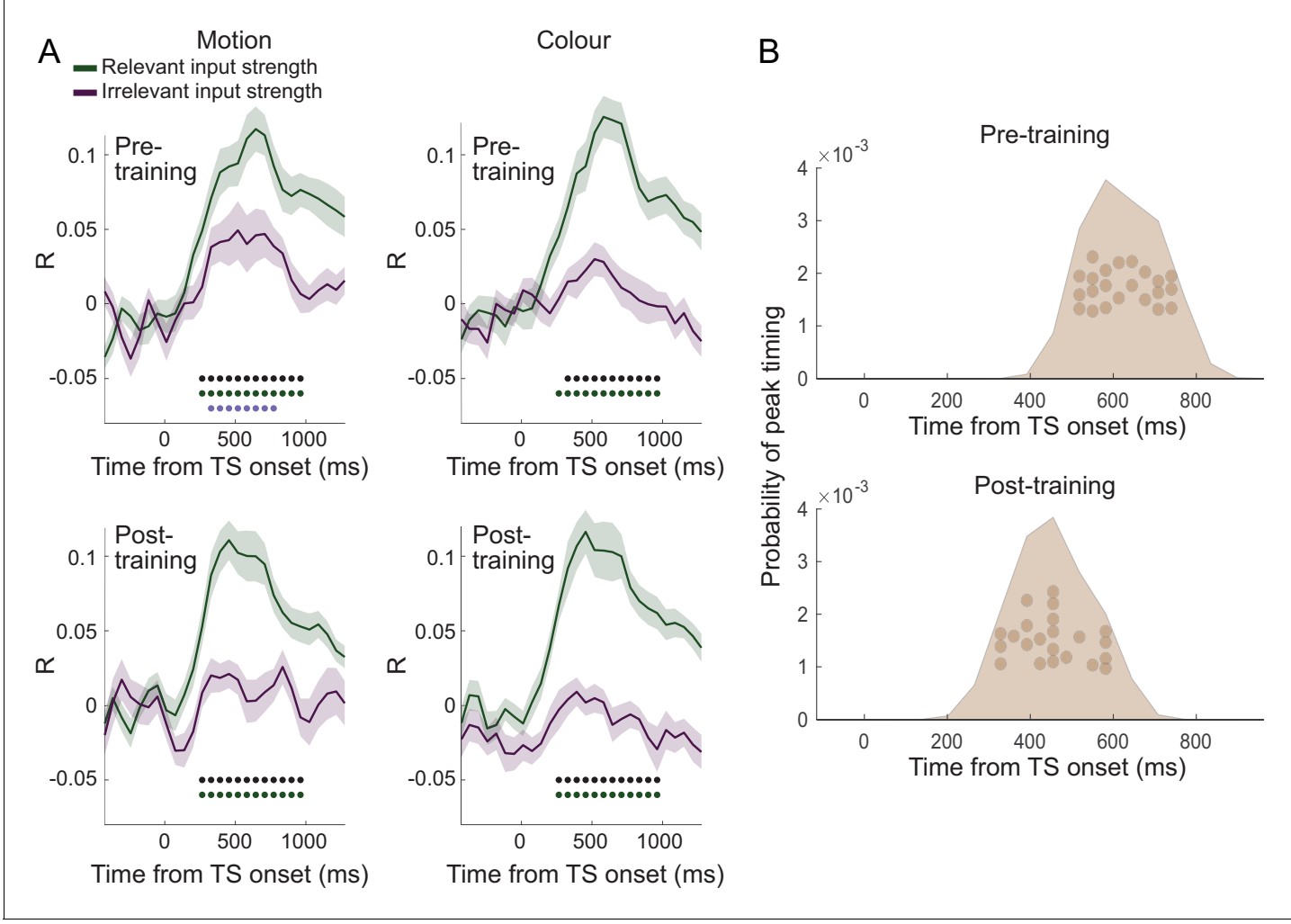

**Figure 6.** Filtering of irrelevant inputs is also present in whole-brain decoding. (**A**) Decoding performed on the source reconstructed signal in 38 parcels covering the whole brain showed a significant difference in decoding accuracy between relevant and irrelevant inputs, suggesting an attenuation of information about irrelevant choice features. Thus, a filtering out of irrelevant information is apparent even when considering data from the whole brain, and it is not affected by extensive training on the task (compare top vs. bottom rows). As in *Figure 3B*, black * denotes significance at p<0.001 for the difference between relevant and irrelevant input coding; green and purple * denotes significance for relevant and irrelevant input decodings, respectively. (**B**) Consistent with the encoding approach focused on PMd, peak decoding times are faster following the training when considering data from the whole brain.

The online version of this article includes the following source data for figure 6:

**Source data 1.** Contains structs of 'pre' and 'post'.
**Source data 2.** Contains 'pre' and 'post' [1 x Subjects].

or task-axes, draws on the observation that electrophysiological responses are suppressed when repeatedly exposed to features to which they are sensitive. In MEG data recorded non-invasively from human participants, each sensor's signal is driven by the summed postsynaptic potentials of millions of neurons. This bulk signal can be manipulated by repeated exposure to a specific feature, causing suppression in the subset of neurons sensitive to this feature and thus reduce the average MEG signal that is measured (*Figure 1*). This repetition approach has greater spatial specificity than multivariate approaches because it manipulates activity in subsets of neurons and can thus be measured within a single sensor, rather than relying on variations in the patterns of activity observed across sensors. While this insight has been exploited for understanding a wide range of cognitive processes, with both fMRI (*Barron et al., 2013*; *Barron et al., 2016b*; *Chong et al., 2008*;

*Doeller et al., 2010*; *Garvert et al., 2017*; *Garvert et al., 2015*; *Jenkins et al., 2008*; *Klein-Flügge et al., 2019*; *Klein-Flügge et al., 2013*; *Piazza et al., 2007*) and MEG/EEG (*Fritsche et al., 2020*; *Henson, 2016*; *Henson et al., 2004*; *Todorovic and de Lange, 2012*), here we extended this framework in one unusual way. As in previous work, using MEG allowed us to measure temporally resolved repetition suppression effects, thus giving insight into precisely when particular features (e. g., input versus choice direction) are processed. This has been exploited previously, for example, to characterize the temporal tuning of repetition suppression (*Fritsche et al., 2020*) or to temporally dissociate stimulus repetition from stimulus expectation effects (*Todorovic and de Lange, 2012*). However, importantly, here we suppressed the signal to multiple different features within the same experiment (colour and motion – when relevant or irrelevant – and motor direction). This can be conceptualized as squashing the neural response onto multiple task-axes, with each feature representing one axis, and allowed us to visualize choice traces in a two-dimensional task space.

The obtained trajectories are comparable to population traces from invasive recordings in NHPs (*Figures 3* and *4*; *Mante et al., 2013*) in that they show a progression from a processing of choice inputs to a processing of the motor response. However, in *Mante et al., 2013*, the influence of input returned to baseline before choice execution. This may be because monkeys were only allowed to indicate a response after a delay, once the motion stimulus had been turned off. By contrast, we allowed participants to respond at any time while the stimulus was presented, and thus the trial could end before input processing returned to zero. Indeed, in our human choice trajectories, input representations started to return to baseline just before the time of response but did not fully return to zero by the time participants responded. Overall, we believe our approach provides an exciting new opportunity, allowing researchers to measure something like the state space trajectories obtained from direct recordings but in tasks that might be difficult for NHPs to do (e.g., involving language), or in human disease.

The neural mechanisms underlying repetition suppression are not fully understood to date. Hypothesized mechanisms include neuronal sharpening, fatigue, and facilitation, with current evidence favouring the fatigue model according to which suppression is caused by attenuated synaptic inputs (*Barron et al., 2016b*; *Grill-Spector et al., 2006*). Another possibility for the observed signal modulations might be related to change detection or prediction-error-like processes caused by novelty or unexpectedness. However, we observe adaptation of not just sensory inputs but also the motor response, and of not just attended but also unattended sensory inputs (*Larsson and Smith, 2012*). This lends support to an interpretation as a suppression rather than a prediction-error-like effect. The temporal dynamics of repetition suppression (e.g., the influence of time-lag) can vary between regions and is likely determined by the neural dynamics and recurrent processing of a given region. Importantly, however, the key effects reported here all come from within the same region. This eliminates inter-regional differences in neural dynamics as a possible explanation for the timing differences we observed between sensory and motor suppression effects. Ultimately, the arbitrary nature of the orientation of the MEG dipoles fitted means that we cannot be certain about whether feature repetition causes a suppression or enhancement in the underlying neural signal. For that reason, we report absolute linear regression coefficients throughout. However, single-unit studies provide strong and consistent evidence for a suppression, rather than enhancement, of neural responses after repeated exposure to a sensory feature (*Barron et al., 2016b*; *Grill-Spector et al., 2006*).

Our behavioural results are consistent with the interpretation that inputs are suppressed as a result of repeated exposure to a feature. If repeated exposure attenuates the received neuronal inputs, thus affecting the processed information content, this might translate into behavioural change because suppressed information cannot be processed as efficiently for making a choice. However, this should only be the case if the relevant sensory dimension is repeated. Our behavioural analyses confirmed this prediction (*Figure 5* and *Figure 5—figure supplement 1*). Repetition of the relevant sensory feature or response, but not the irrelevant sensory feature, reduced choice accuracies, and repetition of relevant, but not irrelevant, sensory features slowed RTs. These results provide further evidence that the repetition suppression approach employed here was effective at manipulating the inputs to the decision process. It seems likely that the same adaptation process that changes the MEG signal in PMd is causing the changes in behaviour we identified. In prior work, choice biases were shown to depend on the precise choice history, and in the majority of participants, choices were biased away from the previous response (*Urai et al., 2019*). This

finding could relate to effects observed here, showing improved performance for response alternation and reduced performance for response repetition.

Our data speak to an important controversy about the mechanisms underlying feature-based, or context-dependent, choice selection. A substantial body of evidence both in human and non-human primates has shown that selection of relevant sensory features occurs through top-down modulations from prefrontal and parietal regions onto early sensory regions (*Buschman and Miller, 2007*; *de Lange et al., 2010*; *Desimone and Duncan, 1995*; *Everling et al., 2002*; *Friston, 2005*; *Friston and Kiebel, 2009*; *Kok et al., 2016*; *Michalareas et al., 2016*; *Moore and Zirnsak, 2017*; *Noudoost et al., 2010*; *Pezzulo and Cisek, 2016*; *Rao and Ballard, 1999*; *Richter et al., 2017*; *Squire et al., 2013*; *Summerfield et al., 2014*; *van Wassenhove et al., 2005*; *Wyart et al., 2015*). By contrast, the study by *Mante et al., 2013* found that irrelevant sensory features were not filtered out but passed forward to the output stage (see also *Siegel et al., 2015*). The authors provided a compelling neural network model that solved feature selection and evidence integration within a single recurrent network. However, the monkeys in *Mante et al., 2013* were extensively over-trained at performing the task and this might have caused the circuits to reorganize. We hypothesized that top-down influences might be more important when someone is naive at doing the task. Indeed, in a task with only relevant features, training changes neuronal responses responsible for interpreting sensory evidence, but not those processing the sensory evidence itself (*Law and Gold, 2010*; *Law and Gold, 2008*). We, therefore, compared participants before and after >20,000 trials of training on the task. To our surprise, the only difference identified between naive and extensively over-trained participants was a shift in peak timings (*Figures 4* and *5*). We did not observe any changes in feature selection. In other words, even following the training, irrelevant features were present but weaker than relevant features in premotor cortex and across the brain. Our data thus suggest partial but incomplete filtering of irrelevant inputs in PMd. This is broadly in line with the effects present in participant's behaviour, where we observed a weak but observable influence of the irrelevant feature (*Figure 5A* and *Figure 5—figure supplement 1A*). Consistent with predictive coding principles and theories of top-down control, the processing of irrelevant inputs was diminished, but partly consistent with *Mante et al., 2013*, they were not completely filtered out. Thus, it seems plausible that some but not all of the feature selection happens at the output stage, in PMd. Overall, the strength of top-down control, or the extent to which task-irrelevant information was filtered out, seemed unaffected by the amount of prior experience on the task.

One important difference between studies in humans and NHPs is the brain area under study. We focused on the premotor area responsible for selecting digit responses, the response modality in our task. By contrast, a large number of studies in NHPs, including *Mante et al., 2013*, record from FEF because choices are indicated using saccades. In terms of their anatomical connections, FEF and PMd can be considered at similar levels of the processing hierarchy for saccadic and digit-motor responses, respectively. FEF has direct projections to the saccade-initiating superior colliculus, while PMd directly projects to the region of M1 that controls hand motor responses. However, based on pyramidal neuron spine count, a good indicator of the hierarchical position of a cortical area, FEF may be at a lower level of the hierarchy than PMd (*Chaudhuri et al., 2015*). Whether or not the positions of PMd and FEF in the processing hierarchy are comparable, this potential difference is unlikely to explain the discrepant findings. Other work in NHPs has shown attentional filtering in regions as 'low' in the processing hierarchy as V4/MT (*Noudoost et al., 2010* and references therein), which would imply sensory information is filtered out before FEF/PMd, consistent with our work.

This leaves outstanding the question as to why equally strong processing of relevant and irrelevant features was observed in NHP state space trajectories (*Mante et al., 2013*), but not in our data. It has been proposed that the site of feature selection may depend on the level of detail afforded by the prediction (*de Lange et al., 2018*; *Hochstein and Ahissar, 2002*). One difference between tasks was that the relevant dimension changed from trial to trial in our experiment but was blocked in *Mante et al., 2013*. There is evidence that FEF shows long-term selection history effects (*Bichot et al., 1996*; *Bichot and Schall, 1999*), which may be promoted by blocking trials. However, *Siegel et al., 2015* do not mention any blocking of trials and report results consistent with *Mante et al., 2013*, making this an unlikely possibility. Recent work has shown attenuation of expected stimulus features when they are attended (*Richter and de Lange, 2019*). However, attenuation of expected information is usually thought to help filter out predictable objects (*Duncan et al., 1997*; *Everling et al., 2002*), for example, via representations of pre-stimulus sensory templates

(*Kok et al., 2017*). Indeed, generally, processing is biased in favour of behaviourally relevant input (for review, see *Desimone and Duncan, 1995*; *Stokes and Duncan, 2014*). There is also a discrepancy in terms of the methods used here and in *Mante et al., 2013*. Mante et al. used multivariate approaches across cells, while we used a repetition suppression manipulation on non-invasive univariate data. However, again, this difference is unlikely to fully account for the discrepancy in findings. A large body of work in NHPs is consistent with our findings but used similar recording, analysis, and training approaches as in *Mante et al., 2013* (*Noudoost et al., 2010*). Furthermore, when we implemented a comparable analysis approach to *Mante et al., 2013*, albeit using the univariate signal of a single sensor and ignoring the repetition suppression manipulation our design was optimized for (*Figure 4—figure supplement 3*), we were able to confirm that signal variation related to sensory stimulus strength was weaker for irrelevant compared to relevant features. Ultimately, the discrepancy between different findings remains to be addressed and highlights a general need for a better understanding of decision-making in environments that require dynamic changes (*Glaze et al., 2015*; *Gold and Stocker, 2017*; *Ossmy et al., 2013*).

Overall, our results reinforce the importance of inter-species translational research, whereby tasks and techniques are used across species, e.g., by using comparable analysis pipelines (*Hunt et al., 2015*; *Kriegeskorte et al., 2008*), by obtaining direct recordings from humans when possible (e.g., neurosurgical patients: *Ekstrom et al., 2003*; *Fell et al., 2001*; *Miller et al., 2013*; *Rutishauser et al., 2010*; *Watrous et al., 2018*), or by recording whole-brain fMRI in NHP species (*Bongioanni et al., 2021*; *Chau et al., 2015*; *Fouragnan et al., 2019*). Our work also emphasizes the importance of developing more mechanistic approaches in human neuroscience, and it shows that the generalizability from NHPs to humans can and should be tested but not assumed (*Passingham, 2009*).

## Materials and methods

### Participants

Twenty-five participants (10 male, 15 female, age range 19–32, mean age 25 ± 0.68) with no history of neurological or psychiatric disorder, with normal or corrected-to-normal vision and who fulfilled screening criteria for undergoing MRI and MEG scanning, took part in this study. The sample size was determined based on the reported sample sizes in previous related studies. One participant dropped out before completing the experiment and two participants' data sets were too noisy even after rigorous data clean-up. The final sample thus included 22 participants (10 male, 12 female, age range 19–32, mean age 25 ± 0.74). There was a problem with processing the MEG data from two sessions (session 2 and session 4 in two different individuals), so only data from three instead of four MEG sessions were included for two participants. The study was approved by the University College London (UCL) Research Ethics Committee (reference 1825/005) and all participants gave written informed consent.

### Experimental procedure

Participants agreed to take part in an initial screening session that ensured that they were safe to undergo MRI and MEG scanning, and that they were able to do the task to a basic standard (e.g., they were not colour blind). No participants were excluded after the screening session. Following the screening, they then took part in a short structural MRI scan and four MEG sessions, two at the beginning and two at the end of the study, spaced 4 weeks apart. They also agreed to complete an hour of training (including short breaks: 1200 trials split into 12 blocks) in the laboratory or on their own computers for 5 or 6 days per week for a total of 22 training sessions spread across 4 weeks (*Figure 4—figure supplement 1A*). Participants who performed their training at home (9 out of 22) agreed to pass on the data to the experimenter on the same day to enable monitoring of progress and to ensure daily completion, and they agreed to perform the task in a quiet environment without interruptions. Personal laptop screens were colour-calibrated to ensure matched stimulus appearance. The four MEG sessions were identical and lasted ~1.5 hr (1024 trials). Participants were reimbursed £250 for their time. Half of the money (£125) was paid out in smaller chunks after each MEG session and each week of behavioural training; the other half was paid upon completion of the entire experiment to discourage drop-out, given the time-intensive nature of this study.

## Experimental task

The task was adapted from *Mante et al., 2013* and contained the same RDM that, in addition to a dominant motion direction, also contained colour information, here varying from predominantly green via grey (neutral) to predominantly red. Each trial contained two sequentially presented coloured RDM stimuli, the AS and the TS. Two separate instruction cues, presented shortly before the AS and TS, signalled whether participants had to judge the direction of motion or the colour dominance of the AS and TS, respectively. This determined the relevant input dimension to focus on. More precisely, within a given trial, the order of presentation was as follows (*Figure 1A*): (1) an instruction cue presented for 150 ms showed a green and red dot to signal that colour was relevant or a left and right arrow to signal that motion was the relevant dimension to attend to for the AS; (2) The AS was presented for 500 ms, either with random motion and 70% colour dominance for green or red, or with non-dominant colour (grey shades) and 70% motion coherence to the left or right; (3) a fixation cross was shown for the 300 ms inter-stimulus interval (ISI); (4) a second instruction cue signalled the relevant dimension for the TS (150 ms); (5) the TS was presented for 500 ms; (5) after another 500 ms of fixation; (6) feedback was presented for 300 ms ('green tick' or 'red cross'). Participants had to respond to both AS and TS using a button press with their right-hand index (left) or middle (right) finger. Because the response to the AS was trivial (dominance level: 70%; accuracy 95 ± 2% during screening), the feedback at the end of the trial related to their response to the TS. TS colour and motion dominance were modulated according to two difficulty levels during the MEG sessions: 12.8 or 25.6%. During the training, we also included two other difficulty levels corresponding to 3.2 and 6.4%.

During the screening session, the four MEG sessions, and the last 3 days of training, this was the precise task used. During the screening session, participants performed six blocks of 128 trials (n = 768 trials) of the task. During MEG sessions, they performed 8 blocks of 128 trials and thus a total of 1024 trials each, allowing a total of 2048 trials from each participant to enter the pre- vs. post-training MEG analyses. During the first 7 days of behavioural testing following the first two MEG scans, a simpler version of the task was used. Participants were only given one stimulus at a time (coherences: 3.2, 6.4, 12.8, 25.6, and 70%) and it only contained either colour or motion ('1-dimensional' stimuli, 1D; *Figure 4—figure supplement 1A*). There was feedback after every stimulus, and even though it was easy to know which feature to attend to (when all dots were grey, it was motion; when they were coloured and static, it was colour), the instruction cue was presented 150 ms prior to stimulus onset. Participants performed 12 blocks of 100 trials (n = 1200) per day, totaling to 8400 trials across the 7 days. Following the 1D task, participants moved on to individual 2D stimuli that simultaneously included colour and motion and performed this 2D task for 12 days (coherences were identical to the 1D task). Again participants performed 12 × 100 = 1200 trials per day, totaling to 14,400 trials of this 2D version of the task. Finally, the last 3 days of training were done on the task described above containing two stimuli presented in quick succession (AS = 70% coherence and TS = 12.8 or 25.6% coherence) and which was identical to the one used during the MEG scans (8 blocks of 128 trials per day and thus 3 × 1024 = 3072 trials in total). Thus, all participants were expected to complete a total of 7 + 12 + 3=22 training sessions. They were told not to take more than 1 day off in a row, but due to sickness, some sessions were missing in some participants (mean number of completed sessions: 21.2). Overall, by the time they came for their third and fourth MEG sessions, participants were expected to have completed 768 (screening) + 2048 (2 MEG) + 8400 (7 days 1D stimuli) + 14,400 (12 days 2D stimuli) + 3072 (3 days full task) = 28,688 trials. Everyone performed the screening, the four MEG sessions, and all seven 1D sessions. Of the 12 2D sessions, participants completed between 5–12 (mean: 10.7) and of the final three full task sessions, they completed between 1–3 (mean: 2.5) before coming back for the two post-training MEG sessions. In total, everyone completed >20,000 trials before the final MEG sessions and on average 26,594 trials (minimum: 20,288, maximum: 28,688).

## Repetition suppression procedure and trial types

All analyses focus on the time of the TS. Importantly, however, the purpose of the AS was to selectively manipulate neurons responding to particular input and response features. For instance, presenting a green AS followed by a predominantly green TS meant that at the time of TS, any MEG sensors influenced by neurons responding to green colour, or by neurons responding to leftward

hand motor responses, should show suppressed responses compared to a situation where a red AS was followed by the same predominantly green TS. In a similar way, we could selectively adapt to green or red colour inputs, right or left motion inputs, and middle/index-finger hand motor responses, and we could do so when a given input was relevant or irrelevant. For example, a green AS followed by a left-motion TS that was predominantly green but while attending to motion, suppressed to green colour when it was irrelevant at the time of TS. Finally, response adaptation could be obtained, e.g., by showing a red AS (leading to a right and thus middle finger response) followed by a right-moving TS (also leading to a response with the middle finger). The full table of conditions can be seen in *Supplementary file 1*. In total, there were 64 conditions: 4 AS x 2 TS contexts (colour/motion) x 2 TS directions (right/left) x 2 TS colours (green/red) x 2 TS coherence levels (12.8 and 25.6%). Trials of all types were interleaved and shown in a random order.

### Stimulus generation

Custom-written MATLAB (The MathWorks, Inc, Natick, MA) code was used to produce a randomized stimulus order for each session and subject, with balanced trial numbers for each of the 64 conditions. For each RDM stimulus, a new random dot placement was generated, and given the appropriate level of coherence and motion. The RDM stimuli were coded using three interleaved streams of stimuli, one per screen refresh rate (17 ms), with the following parameters: speed of dots 4 degrees/second; temporal displacement 50 ms (three screen refreshs); spatial displacement 0.2 degrees/second; unmasked area 10 × 10 degrees; dot diameter 0.3 degrees; and number of dots 100. The stimulus presentation was programmed in MATLAB and performed using the Psychophysics Toolbox (*Brainard, 1997*).

### Behavioural analysis

We recorded choice (left or right button press) and RT to both AS and TS in each trial. To examine training improvements, average RT and **%** correct from the two pre-training MEG sessions were compared with those obtained in the two post-training MEG sessions (which used the same stimuli/schedule; black in *Figure 4—figure supplement 1A*; see also *Figure 4—figure supplement 1B–E*). We used an ANOVA with factors coherence (70% = AS, 25.6% = easy TS, and 12.8% = hard TS), context (colour or motion), and training (pre vs. post) to assess statistical significance (*Figure 4A*).

### MEG and MRI data acquisition

MEG data were recorded continuously at a sampling rate of 600 samples per second using a whole-head 275-channel axial gradiometer system (CTF Omega, VSM MedTech). Participants were seated upright in the scanner and their head location was monitored using three fiducial locations (nasion, left and right pre-auricular points). The distance to the screen was measured to adjust the size of the stimuli and the lights were turned off. Eye movements were recorded (EyeLink software), which required a brief calibration and validation procedure. During each MEG session, participants then performed eight blocks of 7 min of the task, with short breaks in between. Responses were indicated using a keypad with their right-hand index and middle finger. All four MEG sessions (two pre-training and two post-training) were identical in terms of the difficulty, trial structure, and procedure. One of the MEG sessions was followed by a short MRI session, during which a structural T1-weighted MPRAGE scan was acquired on a 3T Magnetom TIM Trio scanner (Siemens, Healthcare, Erlangen, Germany) with 176 slices; slice thickness = 1 mm; TR = 7.92 ms; TE = 2.48 ms; voxel size = 1×1 × 1 mm.

### MEG data preprocessing

MEG data were preprocessed using SPM12 (http://www.fil.ion.ucl.ac.uk/spm/) and custom-written MATLAB code. Data were converted to SPM12 format, a notch filter was applied, and eyeblinks were removed based on the electro-oculogram channel using a regression procedure based on the principal component of the average eye blink as previously explained in *Hunt et al., 2012*. The data were downsampled to 300 Hz, epoched at [−1500,2000] around TS onset, and baseline corrected between [−1500–1100] (thus using a pre-AS baseline). Where necessary, timings were corrected for one frame (1/60 s) between trigger and image refresh, which was based on timings recorded with an in-scanner photodiode. Trials containing artefacts were rejected visually using Fieldtrip's

spm_eeg_ft_artefact_visual. Prior to source localization, data were low-pass filtered at 40 Hz and the blocks from each session were merged.

## MEG source reconstruction

Source reconstruction was performed in SPM12. The structural scans were segmented and normalized to the MNI template. A subject-specific mesh was created using inverse normalization and the three recorded scalp locations were registered to the head model mesh. A forward headmodel was estimated for each session and subject (EEG BEM, single shell). An LCMV beamformer was applied in the window [−250, 750] ms around TS to estimate whole-brain power images on a grid of 5 mm and for source data (virtual timecourse) extraction, using PCA dimensionality reduction to regularize the data covariance estimation (*Woolrich et al., 2011*). Although beamforming has proven to be powerful at reconstructing source signals in electromagnetic imaging, it can be limited in the presence of highly correlated source signals, such as those that can occur bilaterally across hemispheres. To overcome this, a bilateral implementation of the LCMV beamformer was employed, in which the beamformer spatial filtering weights for each dipole were estimated together with the dipole's contralateral counterpart (*Brookes et al., 2007*). Beamformed power images from the two pre-training sessions and the two post-training sessions were smoothed, log-transformed, and averaged, respectively.

## Region of interest

The a priori region of interest for this study was dorsal premotor cortex (PMd). Two analyses performed on our data justified the choice of PMd. First, we ran a broad inclusive beamforming contrast that compared TS trials containing any adaptation (whether colour or motion or response, relevant or irrelevant) with TS trials not containing any adaptation, averaged across pre- and post-training MEG sessions to avoid bias in subsequent analyses. Note that this contrast is entirely balanced for visual TS features and motor responses. For example, equal numbers of right-moving TS trials are on both sides of the contrast. There is a response to each TS and equally many left- and middle-finger responses are present on both sides of the contrast. We identified PMd within the peak cluster of this contrast ($p < 0.05$, familywise error [FWE] cluster-corrected across the whole brain after initial thresholding at $p < 0.001$). Left PMd (x=-37, y=-6, z = 55) was then used for extraction of time courses and further analyses on PMd source data (all subsequent statistical tests were orthogonal to region of interest (ROI) selection). Second, we used an established parcellation that included orthogonalization (to remove spatial leakage between parcels) to extract source data from 38 parcels obtained from an independent component analysis (ICA) decomposition on resting-state functional magnetic resonance imaging data from the Human Connectome Project (*Colclough et al., 2016*; *Colclough et al., 2015*) and confirmed that the strongest task-related effects were present in the parcel that contained left PMd (*Figure 2—figure supplement 1*).

## Linear regression on MEG source data in PMd

We fitted an L2-regularized linear regression (ridge regression) to the raw source data extracted from PMd, which contained the following six regressors capturing task events and repetition suppression effects:

1. Context [1/0; Motion/Colour]
2. Switch instruction [1/0; Switch/No-switch]
3. Relevant input adaptation [1/0; Motion or Colour relevant input adaptation/No relevant input adaptation]
4. Irrelevant input adaptation [1/0; Motion or Colour irrelevant input adaptation/No irrelevant input adaptation]
5. Response adaptation [1/0; Response adaptation/No adaptation]
6. Choice [1/0; Right/Left]

Thus, regressors (3–5) capture the critical repetition suppression manipulation and depend on the preceding AS; regressor 2 captures whether the context changed from the AS to the TS but does not directly relate to repetition suppression; regressors (1,6) capture features related to the TS alone. The regression was applied to each time point around the presentation of the TS ([−500, 1350] ms) for each subject. While some dependencies between regressors were present by design,

the percentage of shared variance was below 0.4 in all cases (Switch with Rel input: $r^2 = -0.34$; Switch with Irrel input: $r^2 = 0.34$; Rel input with Resp: $r^2 = 0.34$). To increase sensitivity, we used a sliding-window approach by averaging time points within 150 ms around the time point and used a step size of 33.3 ms. For each time point, we sub-sampled 90% of trials from all correct trials. We then fitted ridge regression (Matlab's function fitrlinear) to this sub-sample for obtaining linear regression coefficients. Because ridge regression has a hyper-parameter λ (the regularization coefficient), we tuned λ from $\{10^{-5}, 10^{-3}, 10^{-1}, 10^{1}, 10^{3}, 10^{5}\}$ using threefold cross validation for each sub-sample. Specifically, we used the hyper-parameter λ, which performed the best in the threefold cross validation for estimating linear regression coefficients for the subset. We averaged linear regression coefficients across the 10 sub-sets' results for obtaining the estimated linear regression coefficients used for the following statistical analyses. To test whether linear regression coefficients were significant, we generated a null distribution by shuffling the trials within subject. More precisely, we kept rows consistent but shuffled the order of the rows in the design matrix. This preserved the covariance structure of the design matrix in all control shuffles and thus accounted for the possibility that any effects could have been caused by existing correlations between regressors (see also simulation results in *Figure 2—figure supplements 2* and *3*). We generated n = 1000 permutations. We estimated linear regression coefficients using exactly the same procedure as above using these shuffled data.

We used a conservative time window to correct for multiple comparisons across time. Because the context cues came on at −150 ms and responses were on average at around 500 ms, we chose a window of nearly 1 s duration from [−133.3, 950] ms around the TS. At our sampling resolution, this window contained 29 data points, and we corrected all statistical tests on these data across these 29 data samples. This correction was used to establish significance of individual effects (e.g., response adaptation) or differences between two effects (e.g., relevant versus irrelevant input adaptation). Note that, with alpha set at 0.05, and 29 time points, the p-value required for significance would be 0.05/29 = 0.0017 after Bonferroni familywise error rate correction. Therefore, we used a threshold of p<0.001 in the main figures.

We used the absolute values of the linear regression coefficients for statistical analyses and figures because source-localized MEG data have an arbitrary sign as a consequence of the ambiguity of the source polarity. As beamforming is done for each session separately, the sign of the reconstructed dipoles risks being inconsistent across subjects and sessions. We aligned the signs within subject for the two pre-training and the two post-training sessions separately by calculating Pearson's correlation coefficient between average event-related potentials (ERPs) ([−200, 1500]) for sessions 1 and 2, and 3 and 4. In case of a negative correlation, we flipped one session's signals before estimating linear regression coefficients. This ensured that pre- and post-training sessions each used sign-aligned data in a given participant.

To establish whether peak timings between input and response adaptation differed, the peak time for the parameter estimates was established in each participant for both regressors using an out-of-sample procedure. The average peak time across all participants except the left-out participant was determined and the left-out participant's peak was taken as the highest linear regression coefficient in a window of size [−66.7, 66.7] around that group peak. We confirmed that using a wider window of size [−133.6, 133.6] ms did not change our conclusion. This procedure was repeated for all participants and all regressors and peak times were subjected to a one-way repeated-measures ANOVA. We conducted post-hoc pair-wise *t*-test between the peak timings of linear regression coefficients for input and response adaptation. For *Figure 2C,a*, probability distribution of the peak time for each regressor was estimated using the fitdist function in MATLAB for visualization purposes.

Choice traces showing the signal progression in a two-dimensional task space spanned by input and response dimensions to mimic NHP population traces were generated by plotting the linear regression coefficient obtained for input and response adaptation against each other within the same plot.

To investigate training effects on relevant versus irrelevant input processing, we used a two-way ANOVA on the estimated linear regression coefficients with factors training (pre vs. post) and input adaptation (relevant vs. irrelevant) for each time point in the window [−133.3, 950] ms. Bayesian ANOVAs were performed in JASP (JASP Team (2018), https://jasp-stats.org) and were JZS Bayes factor ANOVAs (*Love et al., 2019*; *Rouder et al., 2012*) with default prior scales and enabled

measuring the likelihood of the null hypothesis. Across time points, the largest P(M|data) for the model including the input x training interaction in PMd was P(M|data)=0.166. The winning models were either the Null model (17 time points) or a model with only a factor of input (relevant vs. irrelevant; nine time points), and their P(M|data) was >0.4 across time (mean 0.52).

### Behavioural analysis of RT and accuracy

The six regressors described above were also used in a linear and logistic regression of TS RTs and TS accuracies (1 = correct, 0 = incorrect). For each regressor, a t-test was performed on the linear/ logistic regression coefficients obtained across all participants.

### Task feature processing independent of repetition suppression in PMd

Our experiment was optimized for the analyses described above, which capitalize on the suppression of the MEG signal along multiple task features (or 'axes') induced by repeated exposure (repetition suppression). This is because multivariate approaches, like those performed across neurons in *Mante et al., 2013* would have to be performed across sensors when dealing with MEG data. However, this would not offer the required spatial resolution for studying modulations within a single brain region, here PMd. Nevertheless, in an additional analysis, we tried to implement an analysis approach more similar to the encoding analysis used in *Mante et al., 2013* which did not rely on the repetition suppression manipulation. This analysis was performed across trials on the univariate signal from the virtual PMd sensor. We used the following regressors, which are independent of the adaptation stimulus and only pertain to stimulus and response properties of the TS (*Figure 4—figure supplement 3*):

1. Context [1/0; Motion/Colour]
2. Relevant input strength [–2/–1/+1/+2; indicating the level of positive or negative sensory evidence. The sign of the motion and colour coherence is defined such that positive coherence values correspond to evidence pointing towards a right choice, and negative coherence values correspond to evidence pointing towards a left choice.]
3. Irrelevant input strength [–2/–1/+1/+2; same as the relevant input regressor but pertaining to the sensory information that is irrelevant on a given trial, e.g. colour input strength on a motion trial]
4. Choice direction [1/0; Right/Left]

Therefore, unlike in the first GLM above, in this GLM, regressors are related to stimulus properties of the TS, and unrelated to the AS and the repetition suppression manipulation. The regression was applied to each time point of the time series extracted from PMd using the same fitting and statistical procedures described above (e.g., including regularized regression, multiple comparison correction, extraction of peak timings). We note that Mante et al.'s regressors were identical to ours with the only difference that we merged colour and motion trials for the two input strength regressors to maximize the power of our design.

While we are able to confirm our key effects, there are two important caveats that mean that this analysis is not as sensitive as our main analysis. First, this analysis approach ignores the adaptation stimulus that we know affects the MEG signal at the time of the test stimulus. Second, our experimental design was not optimized for this analysis, and high correlations were present between the relevant input strength and choice direction regressors (r = 0.95). Unlike *Mante et al., 2013* we did not have three levels of coherence and we did not randomize the colour-response mappings, which would have reduced the design correlations. We did not optimize the design for this analysis but instead for a repetition suppression analysis because repetition suppression has higher within-sensor spatial sensitivity.

### Decoding from whole-brain MEG scalp data

Finally, to rule out that we were overlooking potential representations of the irrelevant sensory inputs by focusing solely on PMd, we repeated a similar analysis to the above control analysis on the whole-brain source-reconstructed MEG signal in 38 parcels ('virtual sensors') (*Colclough et al., 2016*; *Colclough et al., 2015*). Again, this analysis did not code regressors in reference to the adaptation stimulus/feature suppression but focused on the properties of the TS. A decoder was constructed separately for each time point around the presentation of the TS ([−506.7,1416.7] ms) for

each subject and each session. To increase sensitivity, we used a sliding-window approach by averaging time points within 150 ms around the time point and used a step size of 63.3 ms. The regressor used to predict current sensory evidence took values from [–2,–1,+1,+2], indicating the level of positive or negative sensory evidence, such that positive coherence values point towards a right choice, and negative coherence values indicate a left choice. Because of the high correlations between sensory evidence and choice direction (see above), we took a conservative approach. First, for each time point and virtual sensor/parcel, choice direction (right or left) was regressed out of the signal. A decoding analysis was then performed on the residuals of each parcel, having accounted for variance explained by choice direction.

For each context (motion and colour), we constructed a decoder for relevant input (e.g., motion input in motion context) and irrelevant input (e.g., motion input in colour context) separately. We used ridge regression, and decoding performance was evaluated using a nested cross validation procedure as follows. We first split all correct trials into 10 sets of trials (tenfold outer-CV). We then split whole trials except one held-out set into three sets of trials (threefold inner-CV). We tuned the hyper-parameter $\lambda$ from $\{10^{-5}, 10^{-3}, 10^{-1}, 10^{1}, 10^{3}, 10^{5}\}$ in this threefold inner-CV. Finally, we fitted the model with the best-performing $\lambda$ from the inner-CV to three sets of trials and obtained the prediction for the one held-out set. This procedure was repeated 10 times. Here, we chose a window of nearly 1 s duration from $[-190, 1036.7]$ ms around the TS for statistical testing. At our sampling resolution, this window contained 18 time points.

Again, Bayesian ANOVAs were performed in JASP. Across time points, the largest P(M|data) for the model including the input x training interaction in the scalp data was P(M|data)=0.273. The winning models were either the Null model (two time points), a model with only a factor of input (relevant vs. irrelevant; 10 time points), or one with timing and input factors but not their interaction (four time points). Their P(M|data) was >0.3 across time (mean 0.56).

## Acknowledgements

YT was funded by Grants-in-Aid for Scientific Research on Innovative Areas from the JSPS (23118001, 23118002) and Uehara Memorial Foundation. MCKF was funded by a Sir Henry Wellcome Fellowship (103184/Z/13/Z). TEJB was funded by Wellcome Senior Research Fellowship (104765/Z/14/Z), Wellcome Principal Research Fellowship (219525/Z/19/Z), JS McDonnell Foundation award (JSMF220020372), and Wellcome Collaborator award (214314/Z/18/Z). We would like to thank Gareth Barnes and Vladimir Litvak for advice on initial data analyses, the whole support team at the FIL for help with data acquisition, and MaryAnn Noonan, Nick Myers, and Lev Tankelevich for helpful discussions on the manuscript.

## Additional information

### Competing interests

Timothy EJ Behrens: Senior/Deputy editor, eLife. The other authors declare that no competing interests exist.

### Funding

| Funder | Grant reference number | Author |
|---|---|---|
| Japan Society for the Promotion of Science | 23118001 | Yu Takagi |
| Uehara Memorial Foundation | | Yu Takagi |
| Wellcome | 103184/Z/13/Z | Miriam C Klein-Flügge |
| Wellcome | 104765/Z/14/Z | Timothy EJ Behrens |
| Wellcome | Principal Research Fellowship 219525/Z/19/Z | Timothy EJ Behrens |
| James S. McDonnell Foundation | JSMF220020372 | Timothy EJ Behrens |

| Wellcome | 214314/Z/18/Z | Timothy EJ Behrens |
| Japan Society for the Promotion of Science | 23118002 | Yu Takagi |

The funders had no role in study design, data collection and interpretation, or the decision to submit the work for publication.

### Author contributions

Yu Takagi, Formal analysis, Visualization, Methodology, Writing - original draft, Writing - review and editing; Laurence Tudor Hunt, Mark W Woolrich, Methodology, Writing - review and editing; Timothy EJ Behrens, Conceptualization, Supervision, Writing - review and editing; Miriam C Klein-Flügge, Conceptualization, Data curation, Formal analysis, Supervision, Validation, Investigation, Visualization, Methodology, Writing - original draft, Project administration, Writing - review and editing

### Author ORCIDs

Yu Takagi (iD) https://orcid.org/0000-0003-0503-785X
Laurence Tudor Hunt (iD) http://orcid.org/0000-0002-8393-8533
Timothy EJ Behrens (iD) https://orcid.org/0000-0003-0048-1177
Miriam C Klein-Flügge (iD) https://orcid.org/0000-0002-5156-9833

### Ethics

Human subjects: The study was approved by the University College London (UCL) Research Ethics Committee (reference 1825/005) and all participants gave written informed consent.

### Decision letter and Author response

Decision letter https://doi.org/10.7554/eLife.60988.sa1
Author response https://doi.org/10.7554/eLife.60988.sa2

## Additional files

### Supplementary files

• Supplementary file 1. Task conditions and the corresponding regressors. The list of task conditions and corresponding regressors of the experiment are shown. The four bold lines are illustrated as examples in *Figure 1B*.

• Transparent reporting form

### Data availability

The code used in the current study and the datasets generated and/or analyzed during the current study are available at the OSF repository (https://doi.org/10.17605/OSF.IO/RJY3Z).

The following dataset was generated:

| Author(s) | Year | Dataset title | Dataset URL | Database and Identifier |
|---|---|---|---|---|
| Takagi Y, Hunt LT, Woolrich MW, Behrens TEJ, Klein-Flügge MC | 2021 | Adapting non-invasive human recordings along multiple task-axes shows unfolding of spontaneous and over-trained choice | https://doi.org/10.17605/OSF.IO/RJY3Z | Open Science Framework, 10.17605/OSF.IO/RJY3Z |

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
