## [Decision Letter]

**Acceptance summary:**

This study used a novel approach that combined measures of human brain activity with high spatial and temporal resolution (using magnetoencephalography, or MEG) and repetition suppression to identify neural representations of task-specific information processing related to the stimulus, task context, and/or motor response during decision-making. The primary finding, which runs counter to many related studies in non-human primates, is that in premotor cortex, neural activity encodes task-relevant features more strongly than task-irrelevant stimuli. The clever approach, and the use of that approach to draw interesting and well-grounded conclusions about information processing in the human brain, were considered particularly noteworthy and likely to inform future studies of human decision-making.

**Decision letter after peer review:**

Thank you for submitting your article "Projections of non-invasive human recordings into state space show unfolding of spontaneous and over-trained choice" for consideration by *eLife*. Your article has been reviewed by 2 peer reviewers, and the evaluation has been overseen by a Reviewing Editor and Joshua Gold as the Senior Editor. The following individual involved in review of your submission has agreed to reveal their identity: Lucas C Parra (Reviewer #2).

The reviewers have discussed the reviews with one another and the Reviewing Editor has drafted this decision to help you prepare a revised submission.

The editors have judged that your manuscript is of interest, but as described below, that extensive revisions are required before it can be considered for publication. The editors and reviewers agree that no further data are required but major conceptual and data-analysis wise clarifications are essential.

We would like to draw your attention to changes in our revision policy that we have made in response to COVID-19 (https://elifesciences.org/articles/57162). First, because many researchers have temporarily lost access to the labs, we will give authors as much time as they need to submit revised manuscripts. We are also offering, if you choose, to post the manuscript to bioRxiv (if it is not already there) along with this decision letter and a formal designation that the manuscript is "in revision at *eLife*". Please let us know if you would like to pursue this option. (If your work is more suitable for medRxiv, you will need to post the preprint yourself, as the mechanisms for us to do so are still in development.)

Summary:

In this paper the authors use a novel technique to disentangle neural representations of distinct choice-relevant variables in MEG data which leverages the phenomenon of repetition suppression. By presenting an 'adaptation stimulus' before each perceptual choice the authors were able to selectively suppress neural activity for different sensory and response modalities and used these suppression signatures to isolate distinct activity components reflecting task context (instruction to attend to motion vs colour), relevant sensory input, irrelevant sensory input, motor response (index vs middle finger) and choice. Repetition suppression was used here with clever experimental design to determine from MEG whether premotor cortex (PMd) "encodes" different properties of the stimulus, task or response during a decision making task. The premise is that if adaptation is observed for a specific feature, then this feature must have been "encoded" in PMd. The main finding is that stimulus, task and response features are all "encoded" in PMd with increasing delay, and that task-irrelevant stimulus properties are "encoded" less strongly. While, prior NHP studies found little difference in the representation of relevant vs irrelevant sensory inputs at the final integration stage, the present study found that irrelevant representations were significantly weaker. A follow-up study in which humans were extensively trained to mimic the task exposure experienced by NHP replicated these results.

The study is interesting on many fronts and relevant to a wide audience. It is, however, essential the revise the manuscript extensively to address several issues including methodological challenges associated with inferring neural computations underpinning decision making from non-invasive recordings and the more general question of the degree to which irrelevant sensory inputs are subject to top-down filtering.

Essential revisions:

1. The rationale for relying on repetition suppression to isolate a neural readout of the decision process needs to be more clearly articulated. Given the emphasis the authors place on comparing their results to previous NHP studies (Mante et al. 2013, Siegel et al. 2015), the authors should explain why they could not have applied a similar decoding approach. The task design means that task context, sensory modality, sensory input strength and choice are already nicely orthogonalised so why is the adaptation step necessary?

2. The query above relates to a more substantive question regarding the degree to which the authors approach can allow us to draw firm conclusions regarding the relative timing with which these distinct variables are represented. For example the authors highlight that sensory representations precede choice representations.

2a. For starters, this is contrary to what Siegel et al. (2015) found – they reported that choice representations emerged before the stimulus even appeared.

2b. More importantly, to what extent can it be assumed that the relative timing of adaptation effects on sensory vs motor components necessarily translates directly to differences in the time at which these variables influence the decision process?

2c. The authors note the well-established fact that stimulus and response repetition is associated with decreased BOLD/EEG/MEG activity in the relevant brain regions but what do we know about the timing of these effects? Can we assume comparable dynamics underpinning sensory and motor adaptation?

2d. For example, recent studies of choice history biases (e.g. Urai et al. 2019, *eLife*) suggest that responses on trial N-1 cause a bias in the rate of evidence accumulation for repeated choices suggesting that the neural dynamics associated with repetition may more complex than a simple attenuation. More pertinently, can we assume that these adaptation/attenuation effects have any impact on the information content for the decision process?

Please clarify these aspects in relevant sections of the manuscript and address with data analyses how repetition impacts sensory encoding versus motor preparation signals.

3a. The contrast between the present study and the aforementioned NHP studies on the point of filtering of irrelevant sensory inputs is striking and interesting. The authors have, however, used a different analysis strategy to that of Siegel and Mante, which could conceivably contribute to this difference. This does not necessarily undermine the novelty and importance of the results but points to some additional possibilities. Please clarify this and consider corroborating the results by implementing a more comparable analysis approach.

3b. The authors could further examine the difference in human versus monkey behaviour. In Siegel et al. (2015), the monkeys exhibited quite strong cross-over effects (ie. RT for motion choices being impacted by stimulus colour). How strong are the cross-over effects in the present study? Please quantify and clarify this issue. This would be helpful to know as it would point to a more fundamental cross-species difference and perhaps rule out the possibility that the cause of the discrepancy lies more in the differences in analysis strategy or neural recording methods.

4. A key difficulty with the narrative of this work is the notion that adaptation=eoncoding. If we have understood correctly, what is actually quantified here is whether a change in experimental condition (from one repeat to the next) drives variance in the PMd signals. Therefore, the analysis treats PMd as a the output of a change detector. But the readout of a change detector does not necessarily need to encode the feature itself. So to claim that presence of adaptation (a weaker response to the stimulus) = good encoding of that stimulus, was found confusing to both reviewers and the reviewing editor. A lot of the language in the result sections equates the two and makes it very hard to parse. Please clarify and justify the rationale.

5. Another central terminological confusion pertains to "Projection into state space" that is in the title and much of the introduction. This gives an impression of multi-variate analysis of MEG data, which is largely not the case in this study. Until Figure 5, all the analysis is on uni-variate neural signals and nothing is "projected", nor is there any use of "subspace" or "decoding" or "encoding". It is clear that the investigators see "adaptation" conceptually as a way to quash neural response in some dimensions, and in that sense the term "projection" may be justified. This is, however, very unusual use of terminology and may be seen confusing by many readers. Please reformulate the title and introduction of the paper to more accurately reflect the content of the paper and better set the expectations for the reader.

---

## [Author Response]

Essential revisions:1. The rationale for relying on repetition suppression to isolate a neural readout of the decision process needs to be more clearly articulated. Given the emphasis the authors place on comparing their results to previous NHP studies (Mante et al. 2013, Siegel et al. 2015), the authors should explain why they could not have applied a similar decoding approach. The task design means that task context, sensory modality, sensory input strength and choice are already nicely orthogonalised so why is the adaptation step necessary?

We agree that both analysis approaches are interesting. However, MEG signals measured non-invasively in humans capture neural activity at a very different spatial scale compared to those obtained from direct recordings in NHPs. This limits the use of multivariate techniques in ways that are not applicable when analysing data from invasive neuronal recordings. To respond to this comment, we will:

1. First explain why, when using MEG, repetition suppression (RS, or “adaptation”) provides the only way to measure the activity of specific neuronal populations within a given brain area. We will explain that multivariate approaches are sensitive to differences across sensors, but importantly will not be able to assess representational differences manifested within a single brain area.

2. Second, we will nevertheless show the results from conducting the suggested analysis on the univariate signal from PMd. Our task was not optimized for this analysis and is less sensitive to distinguish between two of the key variables (relevant sensory inputs and choice direction) in our experiment compared to Mante et al. or Siegel et al. Nevertheless, this new analysis confirms our key findings and has now been included as a supplementary figure.

1. In the study by Mante et al., the decoding approach was successful because it relied upon an analysis approach that looked for multivariate information *across cells*. With MEG, such an approach would have to look across *sensors*, rather than cells, which provides a very different spatial resolution, not sufficient for looking at responses within a single brain region, which was the focus of the present study. Indeed, it has been shown that we may not be sensitive to cellular-level information using multivariate decoding at the level of voxels or sensors when dealing with neuroimaging data (e.g. Dubois…Tsao, 2015). However, RS has been proposed as a technique to target cellular-level information by *experimental design*, rather than by multivariate analysis. As such, it is a univariate rather than a multivariate approach, giving us specificity to neural responses *within* each individual sensor and trial.

This is possible because the adaptation stimulus causes suppression, at the time of the test stimulus, only in *those neurons selective to the specific features* contained in the adaptation stimulus. For example, when the adaptation stimulus is a right-motion stimulus, this means that the MEG signal at the time of the test stimulus presented shortly after is less influenced by the subpopulation of neurons sensitive to right-ward motion. In other words, we can infer the contribution of subpopulations of neurons within a given trial and brain region (or sensor), which multivariate methods do not afford given the spatial scale of MEG. Because Mante and colleagues also studied neural responses within a single brain region, repetition suppression is the closest we can get to their approach using a non-invasive technique.

We have now introduced this rationale more clearly in the Introduction:

“In this study, we focus on choice processes unfolding in dorsal premotor cortex (PMd). PMd is the key region for choosing hand digit responses (Dum and Strick, 2002; Rushworth et al., 2003), the response modality in our choice task. Thus, the neural representations of interest are located within one brain region. Given this spatial focus, repetition suppression provides the best resolution achievable using non-invasive MEG: a single sensor or voxel is sufficient to reveal feature-processing using repetition suppression. By contrast, multivariate approaches rely on spatial patterns detected across sensors, which would not offer the required spatial scale.”

Before conducting the suggested analysis on the univariate PMd signal below, we note that we had already included a multivariate decoding analysis in the original manuscript albeit not specifically for PMd (for the reasons outlined above), but across 38 parcels covering the whole brain (original Figure 5, now Figure 6 in the updated manuscript). We took a conservative approach given high correlations between the regressors coding for choice direction and relevant input strength. Namely, we first regressed out choice direction from the signal in all 38 parcels and then asked whether we could still decode relevant input strength from the residuals. We found that this was possible. We also found that, in spite of the high correlations between the relevant input dimension of choice direction, relevant inputs were coded more strongly than irrelevant inputs across the 38 parcels. We introduced this analysis as a whole brain control to see if irrelevant responses were encoded more strongly anywhere else. Even across all regions, the filtering out of irrelevant inputs was present both before and after training.

2. Nevertheless, as suggested by the reviewers, we have now also conducted an analysis similar to those conducted in the Mante and Siegel papers on the univariate PMd signal. We cannot run decoding on a single timecourse but we can run an encoding-style analysis using comparable regressors to Mante and Siegel, in other words regressors that are unrelated to any repetition suppression effects and instead just code the features of the test stimulus. This can only tell us whether PMd’s univariate signal variance reflects the strength of sensory input or choice direction, but we note that such variation could be caused by other confounding processes, such as attention or choice difficulty. By contrast, the repetition suppression approach captures the precise sensory or motor features that are repeated and is therefore more sensitive (and thus more like the across-cell multivariate approaches employed in Mante and Siegel).

We coded the regressors as

1. Context [-1, 1]

2. Relevant input strength [-2, -1, 1, 2]

3. Irrelevant input strength [-2, -1, 1, 2]

4. Choice direction [-1, 1]

Mante et al.’s regressors were identical to these with the only difference that in addition, they split up the relevant and irrelevant inputs into colour and motion trials. In other words, here under ‘relevant input strength’, we collapse two of their regressors that assess colour input strength in colour trials and motion input strength in motion trials. Under ‘irrelevant input strength’, we collapse across motion input strength in colour trials and colour input strength in motion trials. This seems appropriate to maximise the power of our design given our main question is about the difference between relevant and irrelevant input representations, and not colour and motion inputs, and given we have a smaller number of trials.

However, it is unfortunately not the case that relevant sensory inputs (e.g., right-ward motion) are fully orthogonal to the choice direction (e.g., right button press) in our task (and this would not change if we split up trials into colour and motion trials). At the time of the test stimulus, these two are correlated by *r*=0.95. The two levels of coherence are the only feature that stops them from being fully correlated. In our repetition suppression analysis, the adaptation stimulus (which could for example involve the same-hand response but a different input) was instrumental in helping us to decorrelate these two dimensions.

There are several small but key differences that meant that Mante and Siegel’s experiments were better able to decorrelate these variables. First, Mante and colleagues had three coherence levels in each direction. Because we collected fewer trials in our human participants and included the adaptation stimulus manipulation, we simplified our task and only included two coherence levels. However, as can be seen in the black line in Author response image 1, which shows the correlation between relevant input strength and choice direction at multiple coherence levels, the additional coherence level only helps reduce correlations a little. However, Mante and colleagues introduced a second important manipulation. For trials in the colour context, the location of the saccade targets was randomized, so green random dot-motion stimuli sometimes required a right saccade to a green target located on the righthand side of the screen and sometimes required a left saccade to a green target located on the left-hand side of the screen. This was signalled to the animals using flanker stimuli (which we also used here but kept in fixed locations meaning that colour-direction mappings were fixed across our experiment). This scenario with randomized colour-choice mappings is simulated using the blue line in Author response image 1. Because colour trials made up half the trials, this significantly reduces the correlation between the relevant input strength regressor and the choice direction regressor from >0.9 to <0.5. We note, however, that when splitting up the relevant input strength regressor into motion and colour trials as done in Mante et al., this manipulation only helps to decorrelate one of the two input regressors (colour input strength in colour trials) from the choice direction regressor. The high correlations are still present between the regressor capturing motion input strength in motion trials and choice direction. It would be around ~0.9 even in Mante et al.’s study which is probably why they projected the population response into orthogonal axes before looking at the temporal evolution of input and choice signals. This is what we need repetition suppression for here.

Siegel et al. used 100% coherent stimuli but had seven motion directions and colour levels (see their Figure S1). Their task therefore probed a slightly different type of categorization choice.In summary, not only is our method (MEG) not best suited for conducting multivariate analyses that require the spatial scale of subpopulations of neurons within a single brain region (see first part of the response above), but our task design is also *not* optimized to dissociate relevant sensory information from choice direction at the time of TS, when disregarding the AS manipulation. Because relevant input and choice direction are highly correlated, it is difficult to make inferences about their relative timings. The reason for not including the necessary manipulations to decorrelate these variables was because our experiment was optimised for measuring cellular representations via repetition suppression.

Nevertheless, we conducted the analysis suggested by the reviewers. The results from this new analysis, conducted on the timeseries extracted from PMd (the same one used in the central repetition suppression analyses), confirm our key findings. They show that PMd signals contain information related to all the different task variables, sensory inputs are processed earlier in time compared to motor responses and that irrelevant inputs are represented more weakly than relevant inputs both before and after extensive training (see new Figure 4 —figure supplement 3). One caveat of using this analysis is that we are ignoring the adaptation stimulus which we know will influence activity at the time of the test stimulus.

Overall, we conclude that this is an interesting analysis for drawing parallels between existing non-human primate and human datasets, but not one that our task was optimized for, and not one that gives us the sensitivity that the RS analysis approach can provide. We have included the results from this new analysis as a new supplementary figure (see above), added methodological details to the Methods, and summarize it in a new section in the main text as follows:

Results:

“Task feature processing independent of repetition suppression in PMd

The results described thus far have relied on the use of repetition suppression to manipulate the MEG signals recorded at the time of the TS. […] The strength of sensory inputs is processed earlier in time than the choice direction (motor response), and the strength of irrelevant inputs is processed more weakly than the strength of relevant inputs both before and after extensive training (Figure 4 —figure supplement 3).”

Methods:

“Task feature processing independent of repetition suppression in PMd

Our experiment was optimized for the analyses described above which capitalize on the suppression of the MEG signal along multiple task features (or “axes”) induced by repeated exposure (repetition suppression). […] We did not optimize the design for this analysis but instead for a repetition suppression analysis because repetition suppression has higher within-sensor spatial sensitivity.”

2. The query above relates to a more substantive question regarding the degree to which the authors approach can allow us to draw firm conclusions regarding the relative timing with which these distinct variables are represented. For example the authors highlight that sensory representations precede choice representations.2a. For starters, this is contrary to what Siegel et al. (2015) found – they reported that choice representations emerged before the stimulus even appeared.

We are not sure if we fully follow the inconsistency that is being pointed out. First, our results are consistent, in terms of the observed timings, with those reported by Mante et al. where sensory input representations precede choice representations. This can be appreciated in their population trajectories which deflect in the direction of colour or motion before they begin to deflect in the direction of choice.

Siegel et al. (e.g. in their Figure 1E and 1I, which shows the average spiking activity across all units and brain regions) also find that colour and motion information precedes choice information in terms of latency. This is consistent with the significant timing difference they report: “Motion and color information rose after stimulus onset with a significantly shorter latency for color (98 T 2 ms) as compared with motion (108 +/- 2 ms) information (P < 0.001). Last, choice information rose (193 +/- 1 ms) before the motor responses (270 ms +/- 3 ms) and significantly later than motion and color information (both P < 0.0001).”

We wonder if the reviewers and reviewing editor are referring to the spontaneous pre-trial fluctuations of activity that Siegel et al. report, and which occur before the time of stimulus presentation and predict choice. This is a very interesting result but nevertheless, the strong stimulus-evoked choice activity which we investigated here follows a similar time-course in Siegel et al. to the one we observe in our data and is preceded by the encoding of the sensory properties of the stimulus (e.g. colour and motion).

2b. More importantly, to what extent can it be assumed that the relative timing of adaptation effects on sensory vs motor components necessarily translates directly to differences in the time at which these variables influence the decision process?2c. The authors note the well-established fact that stimulus and response repetition is associated with decreased BOLD/EEG/MEG activity in the relevant brain regions but what do we know about the timing of these effects? Can we assume comparable dynamics underpinning sensory and motor adaptation?

We think these two points (2b/2c) are related and can be answered together.

We agree with the reviewer that there is uncertainty about the relationship between repetition suppression and neuronal representation. However, we are not aware of any additional uncertainty about the timing of repetition suppression and the timing of neuronal representation. Other studies that have examined the timing of RS effects in EEG/MEG have found that the sequential timecourse of RS is consistent with what we know about the timecourse of processing from other techniques (e.g. Stefanics, … Stephan, EJN, 2018; see their Figure 3). Notably, the timings and dynamics observed in our data are exactly what we would have predicted from direct recordings results in Mante et al. And in our response above we have now shown that these timings are similar to timings obtained using an encoding approach. Therefore, we can be confident that the relative timing of adaptation effects translates to differences in the time at which these variables influence the decision process.

We also note that, importantly, all effects of interest come from within the same brain region. This eliminates the possibility that inter-regional differences in adaptation dynamics could be confounding our results. Since the temporal properties of repetition suppression are thought to be determined by the neural dynamics and recurrent processing of a given region, within the same brain region, here PMd, these features can be assumed to be constant.

We think therefore that we can have as much confidence as in any non-invasive study. We have included the following section in the Discussion about the relationship between repetition suppression and neuronal representation:

“The neural mechanisms underlying repetition suppression are not fully understood to date. Hypothesized mechanisms include neuronal sharpening, fatigue and facilitation, with current evidence favouring the fatigue model according to which suppression is caused by attenuated synaptic inputs (Helen C. Barron et al., 2016; Grill-Spector et al., 2006). […] The temporal dynamics of repetition suppression (e.g., the influence of time-lag) can vary between regions and are likely determined by the neural dynamics and recurrent processing of a given region. Importantly, however, the key effects reported here all come from within the same region. This eliminates inter-regional differences in neural dynamics as a possible explanation for the timing differences we observed between sensory and motor suppression effects.”

2d. For example, recent studies of choice history biases (e.g. Urai et al. 2019, eLife) suggest that responses on trial N-1 cause a bias in the rate of evidence accumulation for repeated choices suggesting that the neural dynamics associated with repetition may more complex than a simple attenuation. More pertinently, can we assume that these adaptation/attenuation effects have any impact on the information content for the decision process?

We agree, it is likely that repetition suppression affects the information content. While the mechanisms of repetition suppression are still under debate, there is strong evidence favouring a fatigue model whereby suppression occurs due to an attenuation of the received inputs (see Grill-Spector, Henson, Martin, TICS, 2006; Vidyasager, Parkes, Neuroimage, 2010; Barron, Garvert, Behrens, 2016).

However, if this is the case, and RS affects neural inputs, this manipulation of the neural dynamics is likely to translate into behavioural change as well. We tested if this could be directly shown in a new behavioural analysis. In other words, we tested if, by manipulating subpopulations of neurons via RS, we might as a result have also manipulated the behaviour in specific and predictable ways.

This analysis was conducted on log-RTs and response accuracies (% correct) of the test stimulus choices. We hypothesized that suppression (i.e. repetition) of the relevant feature and of the relevant response finger representation might slow RTs and reduce accuracy (note, however, all t-tests are two-sided) but that irrelevant input suppression should not impact behaviour. In other words, we reasoned that if presentation of e.g., a green adaptation stimulus means that green inputs are attenuated at the time of the test stimulus, we would predict reaction times and accuracy to be affected when green is the relevant feature to attend to at the time of the test stimulus. This is consistent with what we found – the plot in Author response image 2 summarizes these effects.

**Author response image 2. respfig2:** 

RTs: We found that participants were faster to respond to motion than colour stimuli (main effect of context; t(21)=-2.31, P=0.03), and slower when the relevant input had already been processed at the time of the AS and had thus been suppressed/attenuated at the time of the TS (t(21)=6.80, P=1.00e-6). RTs were also slowed by a context-switch (t(21)=7.72, P=1.46e-7) and slower for the left compared to right finger (t(21)=3.06, p=0.016).Accuracy: Participants were less accurate when the relevant input was colour compared to motion (t(21)=2.48, P=0.02), when relevant input was suppressed (t(21)=-8.65, P=2.28e-8), when the response was suppressed (t(21)=-2.28, P=0.03) or when a context-switch was required (t(21)=-9.58, P=4.10e-9).

These data are consistent with the hypothesis that repetition suppression attenuates the processing of information at the level of the synapse (fatigue hypothesis) causing the suppressed information to be processed less efficiently, thus causing slower and less accurate choices. It is likely that the same adaptation process that changes the neural signal at the time of the TS is causing the changes in behaviour identified here. These new results therefore provide additional evidence that the repetition suppression approach was effective at manipulating the inputs to the decision process.

We note that a subset of these behavioural effects was originally presented in Suppl Figure 2E (now Figure 4 —figure supplement 1, for relevant and irrelevant input suppression but not response suppression). However, we had not given it a prominent place in the manuscript. Moreover, the linear/logistic regression analysis above is more consistent with our neural analysis and therefore hopefully more intuitive for the reader to understand.

We now present this result in the main Results section (Behavioural effects predicted by neural adaptation mechanisms), explain the methodological details in the Methods, and have added a new behavioural figure (new Figure 5) and Discussion section summarizing these effects.

“Behavioural effects predicted by neural adaptation mechanisms

We have shown that repeated exposure to sensory inputs or motor responses has an impact on the MEG signal recorded at the time of the TS. […] However, neither RTs nor choice accuracies were affected by repetition of the irrelevant sensory feature (RT: t(21)=0.13, P=0.90, Hedges’ g = 0.04, 95%CI = [-0.57, 0.64]; accuracy: t(21)=0.03, P=0.98, Hedges’ g = 0.01, 95%CI = [-0.60, 0.61]). Other main effects included context (RT: t(21)=-2.31, P=0.03, Hedges’ g = -0.68, 95%CI = [-1.34, -0.06]; accuracy: t(21)=2.48, P=0.02, Hedges’ g = 0.73, 95%CI = [0.10, 1.39]), and context-switch (RT: t(21)=7.72, P=1.46e-7, Hedges’ g = 2.27, 95%CI = [1.40, 3.25]; accuracy: t(21)=-9.58, P=4.10e-9, Hedges’ g = -2.82, 95%CI = [-3.95, -1.82]).”

Finally, we think the relationship to Urai et al.’s findings might be interesting but there are also important differences between our study and theirs. For example, trials lasted more than 5 seconds on average in Urai et al.’s behavioural paradigms, meaning choices were more spread out in time (their Figure 2). By comparison, the interval between the AS and TS choice was only 300ms here (our Figure 1) to facilitate repetition suppression analyses. While repetition suppression effects can still occur after several seconds, they would be significantly weaker as they scale with the time-lag. Nevertheless, there is some overlap and consistency between our findings. Urai et al. focus on the history of the choice made (not the history of sensory inputs received). This is equivalent to the adaptation to the responding hand (‘response adaptation’) we report here which we show above has an impact on behavioural performance. Indeed, Urai et al. find that the most consistent effect is an alternation or shift away from the previous response which could also be interpreted in light of an adaptation effect. Although this is not the interpretation the authors put forward and the focus is specifically on the accumulation or drift-diffusion process, our results are broadly in-line with what they find.

We have added a reference to this study to the results and discussion of the new behavioural results as follows:

“Our behavioural results are consistent with the interpretation that inputs are suppressed as a result of repeated exposure to a feature. If repeated exposure attenuates the received neuronal inputs, thus affecting the processed information content, this might translate into behavioural change because suppressed information cannot be processed as efficiently for making a choice. However, this should only be the case if the relevant sensory dimension is repeated. Our behavioural analyses confirmed this prediction (Figure 5 and Figure 5 —figure supplement 1). Repetition of the relevant sensory feature or response, but not the irrelevant sensory feature, reduced choice accuracies, and repetition of relevant, but not irrelevant, sensory features slowed RTs. These results provide further evidence that the repetition suppression approach employed here was effective at manipulating the inputs to the decision process. It seems likely that the same adaptation process that changes the MEG signal in PMd is causing the changes in behaviour we identified. In other work, choice biases were shown to depend on the precise choice history, and in the majority of participants, choices were biased away from the previous response (Urai et al., 2019). This could relate to effects observed here, showing better performance for response alternation, and worse performance for response repetition.”

Please clarify these aspects in relevant sections of the manuscript and address with data analyses how repetition impacts sensory encoding versus motor preparation signals.

We have addressed these points one by one above and hope the additional analyses have satisfied the reviewer’s and reviewing editor’s concerns.

3a. The contrast between the present study and the aforementioned NHP studies on the point of filtering of irrelevant sensory inputs is striking and interesting. The authors have, however, used a different analysis strategy to that of Siegel and Mante, which could conceivably contribute to this difference. This does not necessarily undermine the novelty and importance of the results but points to some additional possibilities. Please clarify this and consider corroborating the results by implementing a more comparable analysis approach.

As mentioned in reply to point 1, we have now implemented the suggested analysis. We have outlined above why, overall, our data and task design are more suited to a repetition suppression analysis strategy. This is because of the low spatial resolution achieved with MEG, because of high correlations between relevant sensory input and choice direction in a Mantestyle encoding approach, and because the AS would be ignored in such an analysis although we know it has an impact on the MEG signal observed at the time of the TS. Despite these caveats, we can replicate our key conclusions in the suggested analysis: (1) relevant inputs are represented more strongly than irrelevant inputs, both pre- and post-training, (2) all decision variables are processed in PMd and (3) the peak timings suggest a transition from a representation of inputs to a representation of choice, with choice representations emerging slightly after input representations (new Figure 4 —figure supplement 3). There are quantitative differences: timings seem slightly less robust and more variable across individuals in the Mante-style regression analysis; and irrelevant inputs are represented even less strongly in PMd. Both of these are likely due to the less sensitive and less well powered analysis with higher correlations in the design matrix.

As mentioned above, we have added a new paragraph to the main manuscript summarizing these new results, and we have added a new supplementary figure (Figure 4 —figure supplement 3) and refer to it in the main part of the manuscript as corroborating our key conclusions. We have decided to focus on the repetition suppression analysis in the main part of the manuscript because this is the analysis strategy our task was designed and optimized for.

We have also added a few sentences to the discussion to briefly summarize the differences in analysis strategy and data types used here and by Siegel and Mante, leaving open the possibility that these differences might have contributed to the different conclusions drawn. We also note that this discrepancy in findings is present even within the macaque literature where similar direct recording techniques and training regimes are employed. Recordings from several laboratories including Robert Desimone, Tirin Moore, Pascal Fries, John Reynolds, John Maunsell, Stefan True (see review by Noudoost and Moore 2010 and ~20 relevant references therein) show that inputs are filtered via top-down attentional processes, consistent with our findings and other work in humans.

Discussion:

“There is also a discrepancy in terms of the methods used here and in (Mante et al., 2013a). Mante et al. used multivariate approaches across cells, while we used a repetition suppression manipulation on non-invasive univariate data. However, again, this is unlikely to fully account for the discrepancy in findings. A large body of work in NHPs is consistent with our findings but used similar recording, analysis and training approaches as in Mante et al. (Noudoost et al., 2010). Furthermore, when we implemented a comparable analysis approach to (Mante et al., 2013a), albeit using the univariate signal of a single sensor and ignoring the repetition suppression manipulation our design was optimized for (Figure 4 —figure supplement 3), we were able to confirm that signal variation related to sensory stimulus strength was weaker for irrelevant compared to relevant features. Ultimately, the discrepancy between different findings remains to be addressed and highlights a general need for a better understanding of decision-making in environments that require dynamic changes (Glaze et al., 2015; Gold and Stocker, 2017; Ossmy et al., 2013).”

3b. The authors could further examine the difference in human versus monkey behaviour. In Siegel et al. (2015), the monkeys exhibited quite strong cross-over effects (ie. RT for motion choices being impacted by stimulus colour). How strong are the cross-over effects in the present study? Please quantify and clarify this issue. This would be helpful to know as it would point to a more fundamental cross-species difference and perhaps rule out the possibility that the cause of the discrepancy lies more in the differences in analysis strategy or neural recording methods.

We believe the reviewers and reviewing editor are referring to the following effect reported in Siegel et al. (Figure 1D), which was similarly shown in Mante et al. (Figure 1c-f). We had provided an analogous figure in the supplement (previous Suppl Figure 2D):

This shows that cross-over effects look very comparable between the three studies and across species. In our data, cross-over effects were slightly more pronounced pre-training, and for motion influencing colour choices compared to colour influencing motion choices. Importantly, however, the effects do not seem qualitatively different between the two species.

We are not aware of equivalent plots for RTs in Mante or Siegel. Taken together, it does not seem like the difference in behaviour between monkeys and humans is a likely explanation for the differences in the filtering out of irrelevant input observed.

We have now moved the behavioural plot from the supplement to a main figure (new Figure 5) to make this information more easily accessible. We also extended it to include trials with/without adaptation to further illustrate the behavioural impact of repeating a stimulus feature (see question 2d above):

4. A key difficulty with the narrative of this work is the notion that adaptation=eoncoding. If we have understood correctly, what is actually quantified here is whether a change in experimental condition (from one repeat to the next) drives variance in the PMd signals. Therefore, the analysis treats PMd as a the output of a change detector. But the readout of a change detector does not necessarily need to encode the feature itself. So to claim that presence of adaptation (a weaker response to the stimulus) = good encoding of that stimulus, was found confusing to both reviewers and the reviewing editor. A lot of the language in the result sections equates the two and makes it very hard to parse. Please clarify and justify the rationale.

Thank you for this comment which seems important as many readers would probably find the terminology and rationale similarly confusing. Apologies for not conveying our message more clearly.

We think the central question here is whether the difference in the MEG signal when a feature was versus was not repeated is due to (a) a suppression effect seen at the synapses due to a fatigue mechanism as outlined in 1 or (b) a change detection or prediction error (PE) signal caused by novelty or unexpectedness. Maybe repetition leads to a smaller signal because of the absence of a prediction error signalling unexpected change. While this is an interesting question that affects the choice of our terminology and that future work should address, a PE-based explanation would not have a large impact on how we interpret our results.

Nevertheless, we think the fact that we see adaptation of not just sensory inputs but also the motor response, and of not just relevant but also irrelevant sensory features, lends support to an interpretation as a suppression effect rather than a PE-like effect. At least it is difficult to know how a motor suppression effect could be discussed in terms of a prediction error. Therefore, we would prefer to continue discussing our findings in terms of “information encoding”. Nevertheless, we have now removed the term “encoding” from the manuscript and talk about adaptation, neural representations or information processing instead. We have tried to make the rationale clearer from the beginning and we have also added a section to the discussion that talks about the differences between prediction errors and adaptation.

Discussion:

“The neural mechanisms underlying repetition suppression are not fully understood to date. Hypothesized mechanisms include neuronal sharpening, fatigue and facilitation, with current evidence favouring the fatigue model according to which suppression is caused by attenuated synaptic inputs (Helen C. Barron et al., 2016; Grill-Spector et al., 2006). Another possibility for the observed signal modulations might related to change detection or prediction-error-like processes caused by novelty or unexpectedness. However, we observe adaptation of not just sensory inputs but also the motor response, and of not just attended but also unattended sensory inputs (Larsson and Smith, 2012). This lends support to an interpretation as a suppression rather than a prediction-error-like effect.”

5. Another central terminological confusion pertains to "Projection into state space" that is in the title and much of the introduction. This gives an impression of multi-variate analysis of MEG data, which is largely not the case in this study. Until Figure 5, all the analysis is on uni-variate neural signals and nothing is "projected", nor is there any use of "subspace" or "decoding" or "encoding". It is clear that the investigators see "adaptation" conceptually as a way to quash neural response in some dimensions, and in that sense the term "projection" may be justified. This is, however, very unusual use of terminology and may be seen confusing by many readers. Please reformulate the title and introduction of the paper to more accurately reflect the content of the paper and better set the expectations for the reader.

Thank you – we have changed the title and introduction as requested. It now says:

Title: “Adapting non-invasive human recordings along multiple task-axes shows unfolding of spontaneous and over-trained choice”

Introduction:

“Here, we extend the repetition suppression framework in one crucial way: we suppress the MEG signal to multiple different features within the same experiment. Adaptation along each feature can be conceptualised as “squashing” the neural response along one task dimension, or task axis. This assimilates task axes in multi-dimensional state space derived from recording many neurons, but using an experimental manipulation of a univariate, rather than multivariate signal. Thus, we ask whether repetition suppression along multiple features can mimic projections onto multiple task axes. If so, this would be the closest we can get to measuring multiple cellular-level representations within a single brain region in humans using MEG, with temporal resolution in the order of milliseconds thanks to the temporal precision of MEG.”